

# A survey on multi-lingual offensive language detection

Khouloud Mnassri, Reza Farahbakhsh, Razieh Chalehchaleh, Praboda Rajapaksha, Amir Reza Jafari, Guanlin Li and Noel Crespi

Samovar, Telecom SudParis, Institut Polytechnique de Paris, Palaiseau, France

## ABSTRACT

The prevalence of offensive content on online communication and social media platforms is growing more and more common, which makes its detection difficult, especially in multilingual settings. The term "Offensive Language" encompasses a wide range of expressions, including various forms of hate speech and aggressive content. Therefore, exploring multilingual offensive content, that goes beyond a single language, focus and represents more linguistic diversities and cultural factors. By exploring multilingual offensive content, we can broaden our understanding and effectively combat the widespread global impact of offensive language. This survey examines the existing state of multilingual offensive language detection, including a comprehensive analysis on previous multilingual approaches, and existing datasets, as well as provides resources in the field. We also explore the related community challenges on this task, which include technical, cultural, and linguistic ones, as well as their limitations. Furthermore, in this survey we propose several potential future directions toward more efficient solutions for multilingual offensive language detection, enabling safer digital communication environment worldwide.

## INTRODUCTION

Online offensive language has become an increasingly prevalent issue that has widespread concern among policymakers, civil society organizations, and even the general public. The extended use of social media platforms is an important facilitator to express offensive and hateful content, especially with the anonymity provided, and to disseminate them widely to a global audience. This has led to a rise in this content, which has had serious negative consequences for many individuals and communities from various demographics. Given the rise in the prevalence of offensive language on popular social media platforms such as Twitter and Facebook, there has been a growing number of proposed techniques for identifying them specifically in the monolingual (*i.e.*, English) setting. Although research on the multilingual dimension of offensive language is a relatively recent area of research, numerous studies have attempted to address this issue comprehensively. The detection of this content helps to identify patterns and trends across different languages and cultures, allowing for a better understanding of the underlying factors that contribute to offensive language in different contexts. Knowledge gained through this can be used to develop targeted interventions to address hate speech more effectively.

Corresponding author
Khouloud Mnassri, khouloud.
mnassri@telecom-sudparis.eu

This survey provides a comprehensive overview of the detection of offensive language and more specifically hate speech in a multilingual setting using various techniques. It examines early research and cutting-edge approaches, highlighting gaps and improvements in multilingual and cross-lingual existing models. The study carefully analyzes multilingual datasets, outlines global initiatives and projects combating offensive language, and underscores the importance of readily available open-source tools in fostering further research and practical applications. Finally, it assesses current challenges and potential solutions, aiming to set a future research direction in multilingual offensive language detection.

While there has been a number of survey papers on hate speech detection in general, our survey paper is among the first few studies to provide a comprehensive summary of how the problem is addressed in multilingual scenarios including an extensive summary of the datasets used. Moreover, this article highlights significant resources such as projects, products, and APIs that are essential for analyzing multilingual hatred and offensive content. Finally, we discuss the challenges and limitations of the existing techniques and potential future works.

## Motivation and impact

Based on the latest research, there are various compelling reasons for investigating and scrutinizing hateful content on social media platforms, with the ultimate goal of fostering a secure, considerate, and diverse online community. Such motivations include (i) detection and the mitigation of harmful behavioural patterns by analysing the trends of hateful content (*Al-Hassan & Al-Dossari, 2019*), (ii) preserving a safe online community by analysing and diminishing offensive, insincere and unsafe content from the social media platforms (*d'Sa, Illina & Fohr, 2020*), (iii) analyzing and safeguarding marginalized user communities that are targeted by hate speech and abusive content due to their race, ethnicity, gender, sexuality, religion, or other identifiable traits (*Al-Hassan & Al-Dossari, 2019*), and most importantly (iv) ensuring adherence to legal and regulatory frameworks in multiple countries that prohibit the propagation of hate speech and other forms of harmful content (*Bakalis & Hornle, 2021*). As a result, analyzing hateful and abusive content can contribute to a more positive user experience on social media platforms by providing a safer and more respectful online community, as well as support efforts to combat online hate speech and provide insight into real user behaviours.

## Definition of hate speech and offensive language

One primary challenge in recognizing content as offensive or not is that, up to date there is no widely acknowledged unique definition of offensive and hate speech. This is primarily due to the ambiguous, subjective, and personal interpretations of whether a speech is "hatred" or expresses "offensive" (*Fortuna, Soler & Wanner, 2020*). Unspecified definitions can result in increased subjectivity in annotations which then facilitates generating a biased model (*Davidson, Bhattacharya & Weber, 2019*).

As shown in Fig. 1, due to the shared characteristics and effects, hate speech and aggressive content can be categorized as types of offensive content (*Poletto et al., 2021*).

**Figure 1** Categories of offensive content: offensive content, hate speech, and aggressive content are distinct but they are often overlapping categories that involve harmful or negative expressions. While there is some overlap between these categories, their distinctions lie in their specific intentions, targets, and impacts on individuals or communities.

Offensive language can encompass a broader range of content that may cause discomfort or offense, while hate speech is specifically targeted at marginalized communities or individuals, aiming to spread prejudice, hostility, or discrimination, and subject to legal and social consequences. Researchers primarily focus on analyzing hate speech rather than general offensive content detection due to its severity of harm and the significant legal and policy implications it carries in many countries. Compared with the law-enforcing based hate-speech definitions such as the descriptions provided by, the Europe union commission (*Wigand & Voin, 2017*), the European Union Agency for Fundamental Rights (https://fra.europa.eu) and the UN Human Rights Office (https://www.ohchr.org), social media platforms should consider a broader definition for the hatred and offensive content.

Generally, hate speech in the context of social media commonly refers to any content that discriminates against or attacks individuals or groups based on several characteristics such as race, ethnicity, religion, sexual orientation, gender, or other identifiable features. This definition often aligns with community policies and content moderation guidelines of various social media platforms, drawing from assigned legal frameworks, academic research, and societal norms concerning discriminative behavior and hateful content online.

## Multilingual *vs.* cross-lingual

In NLP, both multi-lingual and cross-lingual approaches deal with processing and understanding languages, which usually are different from English. However, there are differences in the way of using these terms:

Multi-lingual NLP: Multi-lingual NLP refers to the development and application of NLP models and techniques that can handle multiple languages simultaneously. This involves creating models that can process and understand different languages without the

need for language-specific models. Multi-lingual models are trained on data from multiple languages and learn to capture shared linguistic patterns and representations across languages. These models can perform tasks such as text classification, named entity recognition, or machine translation across multiple languages. They are designed to provide a generalized solution for various languages, but they may not achieve the same level of performance as language-specific models.

Cross-lingual NLP: Cross-lingual NLP focuses on enabling communication and understanding between different languages. It involves developing techniques to transfer knowledge, resources, or models from a resource-rich language (often referred to as a source language) to resource-poor languages (target languages). The goal is to leverage the knowledge and resources available for one language to enhance NLP tasks in another language. Cross-lingual approaches can include tasks such as cross-lingual document classification, cross-lingual information retrieval, or cross-lingual word embeddings. Techniques used in cross-lingual NLP include machine translation, word alignment, parallel corpora, and bilingual dictionaries (*Pikuliak, Simko & Bieliková, 2021*).

## Structure of this survey

Our survey incorporates an extensive exploration in existing approaches and datasets of multilingual hate speech and offensive language detection. The structure of our review is as follows: In 'Background on Multilingual Hate Speech Phenomena', we provide an examination of the previous surveys on hate speech detection. We focus on the studies on multilingual aspect of this task, where we carefully study their deficiencies in order to fill this gap in our survey. Next, "Approaches on Multilingual Hate Speech Detection" presents a thorough review of the existing approaches used for multilingual and cross-lingual offensive language detection. We give interpretations from our findings and summarize these studies in a comprehensive table. Following that, we examined the available resources in this field, starting in "Datasets on Multilingual Hate Speech Detection" with a detailed review of available multilingual datasets by a deep analysis of these corpora introducing their languages and main topics. Next, in "Resources for Multilingual Hate Speech Detection" we illustrate the different international collaborative projects provided for multilingual hate speech detection. We also present community challenges and competitions that focused this task. We, then, introduce a variety of publicly available source codes and APIs. Lastly, in "Challenges and limitations", we present the challenges encountered in the field of multilingual hate speech detection. We also emphasize the limitations, as well as a set of future directions that, we believe, will help to overcome these obstacles and to progress further in the task of multilingual hate speech detection.

**Who can benefit from this survey?** This survey aims to serve as a pivotal roadmap for both the research community and business sector, delving into the current landscape and future direction of multilingual offensive language detection. For researchers in the academic field, it serves as a comprehensive synthesis, describing the ongoing advancements, evolving approaches, and available resources, including datasets, within

this domain. It carefully discusses challenges, and limitations, and proposes promising directions for future exploration. In the business sector, this survey offers valuable understanding, serving as a guide for decision-makers. It helps in the assessment of the implementation of multilingual offensive language detection moderators for online textual content, particularly on social media platforms.

In order to give an in depth understanding of the state of the multilingual offensive language detection field, we provide an extensive review in this article. We carefully explore a number of aspects: mainly current approaches and a wide range of datasets that include low-resource languages, and also, available resources like collaborative projects and tools, challenges faced and possible recommendations for future development. As a result, academics as well as professionals can benefit considerably from our survey. This study gives significant recommendations and defines solutions for future directions, especially proposing getting benefit from the recently released LLMs and generative pre-trained models, therefore, contributing to the compilation of existing knowledge and supporting ongoing progress in reducing hate speech across languages and groups.

## BACKGROUND ON MULTILINGUAL HATE SPEECH PHENOMENA

### Previous surveys

Several previous studies have given comprehensive analysis of hate speech detection in different aspects, focusing on presenting to the community the related data, approaches and multilingual methods, and existing products. In this section, we aim to explore these surveys paying more attention to the multilingual aspect of the field. As displayed in Table 1, some of the existing survey studies focus on presenting definitions and notions related to the domain, as well as an examination of current approaches like in *Poletto et al. (2021)*, *Pamungkas, Basile & Patti (2021b)*, *Chhabra & Vishwakarma (2023)*. Meanwhile, other surveys focus on introducing the available sources of the topic, such as data and available source-code (*Vidgen & Derczynski, 2020*; *Poletto et al., 2021*; *Schmidt & Wiegand, 2017*).

Table 1 illustrates previous literature reviews in the field of hate speech detection in multilingual and in general settings. We have carefully studied and analyzed these surveys, which enabled us to construct a comprehensive narrative review in order to cover the deficiencies in these surveys. More specifically, we aim to give an overview of the existing multilingual and cross-lingual approaches, similar to *Schmidt & Wiegand (2017)*, *Yin & Zubiaga (2021)*, *Fortuna & Nunes (2018)* (on English data), to *Pamungkas, Basile & Patti (2021b)* focusing on cross-lingual methods, as well as *Chhabra & Vishwakarma (2023)*, where they basically displayed monolingual approaches in some specific languages. Adding to that, some previous surveys have presented existing corpora in the domain, in some specific languages as in *Poletto et al. (2021)*, *Jahan & Oussalah (2021)*, *Chhabra & Vishwakarma (2023)*, and more widely in *Pamungkas, Basile & Patti (2021b)* presenting datasets in 18 different languages, and in *Vidgen & Derczynski (2020)*, providing an open source website to 63 datasets in multiple languages (https://hatespeechdata.com/). One metric aspect, to take into consideration to define our survey type is based on the forms of

**Table 1 Key previous surveys on the topic of (multilingual) hate speech detection.**

| Title | Main focus of the survey | How to differentiate it with our survey | Year | Type* |
|---|---|---|---|---|
| A survey on hate speech detection using natural language processing (*Schmidt & Wiegand, 2017*) | Investigating automated identification of hate speech through NLP, using linguistic and semantic features. Highlighting how identifying user profiles involved in spreading hateful content, and presenting supervised and semi-supervised approaches on English data. | No focus on multilingual aspect of hate speech detection, only on studies conducted on English datasets. | 2017 | Narrative |
| Towards generalisable hate speech detection: a review on obstacles and solutions (*Yin & Zubiaga, 2021*) | Presenting NLP methods used for automated hate speech detection on online social media networks. | Not presenting multilingual hate speech detection | 2021 | Narrative |
| A survey on automatic detection of hate speech in text (*Fortuna & Nunes, 2018*) | Provide an overview of the researches conducted in hate speech detection, which includes describing available methods and resources. | General overview about hate speech detection, not focusing on the multilingual aspect of the subject. | 2018 | Systematic |
| A systematic review of hate speech automatic detection using natural language processing (*Jahan & Oussalah, 2021*) | Focusing on the use of deep learning technologies and architectures in hate speech detection, with emphasis the sequence of pipeline processing. | Although presenting some resources (datasets and some available Github projects) in different languages, but no detailed overview on multilinguality. | 2023 | Systematic |
| Surveys on multilingual hate speech | | | | |
| Directions in abusive language training data, a systematic review: Garbage in, garbage out (*Vidgen & Derczynski, 2020*) | Comprehensive review of 63 abusive language datasets in several languages. It addresses the opportunities and problems of open science in this area and provides experts building new abusive content databases. | Only focus on datasets, not considering existing multilingual approaches in the field. | 2020 | Systematic |
| Resources and benchmark corpora for hate speech detection: a systematic review (*Poletto et al., 2021*) | Analyzing the annotated collections of texts released by the broader community, considering their method of creation, topic, language range, and other pertinent factors. | No analysis of the existing methods used in multilingual hate speech detection. | 2021 | Systematic |
| Towards multidomain and multilingual abusive language detection: a survey (*Pamungkas, Basile & Patti, 2021b*) | A study of existing researches about the available datasets and methods used in cross-domain and cross-lingual cases. | Focus on cross-lingual side only in the hate speech detection. No analysis on the available products or resources in the community, used and can be used in multilingual detection of hate speech. | 2023 | Narrative |
| A literature survey on multimodal and multilingual automatic hate speech identification (*Chhabra & Vishwakarma, 2023*) | A survey of hate speech identification methods (strengths and weaknesses), and popular benchmark datasets. | Presenting approaches in several languages (monolingual), but no focus on multilingual nor cross-lingual approaches. | 2023 | Narrative |

Note:
* A narrative review is a more subjective and qualitative study used to create a story from the literature in order to summarize the findings. In contrast, a systematic review is more objective and quantitative, used to discover and evaluate the available literature in order to address a certain research topic (*Hammersley, 2001*).

literature reviews: *Narrative* and *Systematic* reviews. In fact, when writing a narrative review from the literature findings, authors are utilizing a more subjective and qualitative research technique. However, they employ methodical/systematic research methodology using a more quantitative and objective approach when working on systematic reviews. These reviews are considered as a link between practice or policy-making and research. Narrative reviews are also noticed to have this linking function; however, they are usually utilized in order to handle more general and complex subjects (*Hammersley, 2001*).

## Surveys on hate speech—from general perspective

Several surveys have studied hate speech and offensive language detection. In 2017, *Schmidt & Wiegand (2017)* summarized the primary NLP aspects of automated hate speech detection: illustrating the various types of feature representation and, the existing supervised and semi-supervised techniques and their limitations. The authors, proposed as future direction, the urge to analyze hate speech detection from a multilingual viewpoint. Adding to that, *Yin & Zubiaga (2021)* described the most commonly implemented approaches in the field, such as dictionaries, bag-of-words, N-grams, among others. Moreover, *Fortuna & Nunes (2018)* presented a systematic overview. They recap the various approaches and resources available in this domain. They also critically examined valuable resources like datasets, illustrating some of the existings in different languages (English, Dutch and German). Another systematic survey has been conducted in 2021 (then updated in 2023), where *Jahan & Oussalah (2021)* addressed thorough aspects including the language used, the pipeline processing, and the techniques used, focusing primarily on deep learning approaches. They also presented many sources in some languages (data and GitHub projects).

## Surveys on multilingual hate speech—multilingual perspective

Multilinguality is getting more popular in the task of hate speech detection and there are recently some surveys on this aspect, covering existing approaches and resources. These studies pay more attention to the considerable variations in the existing studies that aim to cover other languages (other than English), as well as more concepts related to offensive language (Racism, sexism, among others). This is required to build more generalized approaches. In 2021, *Poletto et al. (2021)* comprehensively studied the annotated datasets of hate speech, taking into account the creation process, subject case, language coverage, and other pertinent factors about the existing lexica and benchmark datasets in different languages. Moreover, few overviews have been conducted on the topic of multilingual offensive language detection, where the survey of *Pamungkas, Basile & Patti (2021b)* presented the approaches and the available corpora employed in cross-domain and cross-language techniques. Moreover, the survey of *Chhabra & Vishwakarma (2023)* provided a comprehensive review of hate speech definitions, exploring the essential textual analysis procedures used. The survey also described the advantages and disadvantages of multimodal and cross-lingual approaches.

## Existing gaps in the previous surveys

Although several review studies have been written on the task of hate speech detection, but still there are several aspects that are not covered in those studies especially when it comes to presenting an in-depth comprehension of the multilingual aspect of this area. Our study is a narrative review that seeks to close this gap by exploring larger number of characteristics of multilingual offensive language detection: from existing approaches and available datasets to related collaborative projects and resource products that include community challenges, source codes, and APIs. Adding to that, this study will also analyze the associated challenges and limitations in the field. Furthermore, we aim to get into

future research directions providing a roadmap for the research progress in multilingual hate speech detection.

# APPROACHES ON MULTILINGUAL HATE SPEECH DETECTION

## Existing approaches

The value of studying multilingual offensive language detection has earned attention in recent years. This increasing interest is a consequence of the linguistic variety within social media platforms. The availability of multilingual datasets, especially from social media platforms that are used worldwide, has made it possible to develop algorithms to detect this content in various languages. Some research studies used to focus on creating monolingual models, working on languages other than English, but they have later evolved into cross-lingual and multilingual approaches, utilizing rich resource languages in order to detect the offensive language in low-resource ones totally unseen using zero-shot learning, or, using few-shot learning (*Goodfellow, Bengio & Courville, 2016*).

This section aims to provide an overview of existing approaches for this task. To that end, we organize the existing studies into eight distinct groups (as shown in the first column of Table 2): Traditional Machine Learning (where we found logistic regression (LR) models), deep neural networks (DNN), transfer learning (TL), machine translation (MT), ensemble learning (EL), meta learning (Meta-L), multitask learning (Multitask-L), and unsupervised learning (UL). By categorizing these approaches, we make it possible to analyze them carefully and present an overview of the evolution of methods used to tackle multilingual and cross-lingual offensive language detection, as described in the next subsection 'Analysis of the existing approaches'.

As for languages presentation, we will use ISO 639-2 codes (https://www.loc.gov/standards/iso639-2/php/code_list.php).

**Methodology of research:** Our research of these existing approaches is based on information from earlier pertinent surveys (mentioned in the previous section). We also used specific keywords, such as "multilingual/cross-lingual offensive language detection", "multilingual/cross-lingual hate speech detection", "multilingual/cross-lingual abusive language detection", among others, to find studies that were published in IEEE Xplore, ACM Digital Library, Google Scholar, among others. By focusing on publications that were released in 2019 and beyond (until July 2023), we ensured that our survey included the most cutting-edge approaches for multilingual and cross-lingual offensive language detection. Moreover, we won't cover monolingual approahces in our study about existing approaches since our focus is basically directed to multilingual approaches, however, we mention some of the most relevant ones we found in 'Other Technologies' subsection. A summary of the identified approaches are presented in Table 2 and each of the eight techniques are detailed in the following part.

### Logistic regression (LR)

There aren't many machine learning-based approaches for detecting multilingual offensive language. In fact, deep neural networks and transfer learning-based methods have shown

**Table 2** Overview of approaches on multilingual and cross-lingual hate speech detection.

| Techniques | Ref. ◇/♣ | Focused Languages | Approach (feature extraction methods) | Year |
|---|---|---|---|---|
| Logistic regression (LR) | Vashistha & Zubiaga (2021) ◇ | Hi, En and Code Mixed | Word embedding for feature extraction | 2020 |
| | Aluru et al. (2020) ◇ | Ar, En, De, Id, It, Pl, Pt, Es and Fr | MUSE and LASER for feature extraction | 2020 |
| Deep neural network (DNN) | Vashistha & Zubiaga (2021) ◇ | Hi, En and Code Mixed | CNN-LSTM (Word embedding) | 2020 |
| | Elouali, Elberrichi & Elouali (2020) ◇ | Ar, It, Pt, Id, En, De, Hi-En Code Mixed | CNN (Character-level representation) | 2020 |
| | Jiang & Zubiaga (2021) ♣ | En, Es and It: 6 languages pairs | Bi-LSTM based capsule network (FastText) | 2021 |
| Transfer learning (TL) | Vashistha & Zubiaga (2021) ◇ | Hi, En and code mixed | BERT | 2020 |
| | Aluru et al. (2020) ◇ | Ar, En, De, Id, It, Pl, Pt, Es, Fr | mBERT | 2020 |
| | Wang et al. (2020) ◇ | En, Tr, Da, El and Ar | XLM-R | 2020 |
| | Bhatia et al. (2021) ◇ | En, Hi and Mr | XLM-R, mBERT, DistilmBERT (emoji2vec) | 2021 |
| | Roy et al. (2021a) ◇ | En, De, Hi | XLM-R | 2021 |
| | Deshpande, Farris & Kumar (2022) ◇ | En, Ar, De, Id, It, Pt, Es, Fr, Tr, Da and Hi | mBERT (MUSE and LASER) | 2022 |
| | zahra El-Alami, Ouatik El Alaoui & En Nahnahi (2022) ◇ | En and Ar | BERT, mBERT and AraBERT | 2022 |
| | Ghadery & Moens (2020) ♣ | En, Da, El, Ar and Tr | mBERT | 2020 |
| | Ranasinghe & Zampieri (2020) ♣ | En, Hi, Bn and Es | XLM-R | 2020 |
| | Dadu & Pant (2020) ♣ | En, El, Da, Ar and Tr | XLM-R | 2020 |
| | Stappen, Brunn & Schuller (2020) ♣ | En to Es | XLM-R based AXEL | 2020 |
| | Ranasinghe & Zampieri (2021b) ♣ | En, Ar, Bn, Da, El, Hi, Es, and Tr | XLM-R | 2021 |
| | Pelicon et al. (2021a) ♣ | En, Es, De, Id and Ar | mBERT, LASER | 2021 |
| | Tita & Zubiaga (2021) ♣ | En, Fr | mBERT, XLM-R | 2021 |
| | Ranasinghe & Zampieri (2021a) ♣ | En and 6 Indian languages: Indo-Aryan (Bn, Hi-En, Ur-En) and Dravidian (Kn-En, Malayalam-En, Ta-En) | mBERT, XLM-R | 2021 |
| | Pelicon et al. (2021b) ♣ | Ar, Hr, De, En, and Sl | mBERT, CseBERT | 2021 |
| | Vitiugin, Senarath & Purohit, 2021 ♣ | En and Es | MLIAN: Multilingual Interactive Attention Network (LASER, DistilmBERT) | 2021 |
| | Eronen et al. (2022) ♣ | En, De, Da, Pl, Ru, Ja and Ko | mBERT, XLM-R | 2022 |
| | Zia et al. (2022) ♣ | En, Es, It, De, Ar, El and Tr | RoBERTa, BERT | 2022 |
| Machine translation (MT) | Ibrohim & Budi (2019b) ♣ | Hi, En, and Id | Google Translate API to translate all data between source and target languages. | 2019 |
| | Aluru et al. (2020) ♣ | Ar, En, De, Id, It, Pl, Pt, Es and Fr | Google Translate API to translate all the datasets in different languages to English = input to BERT | 2020 |
| | Jiang & Zubiaga (2021) ♣ | En, Es and It: 6 languages pairs | Google Translate API to translate all data between source and target languages | 2021 |

(Continued)

| Techniques | Ref. ◇/♣ | Focused Languages | Approach (feature extraction methods) | Year |
|---|---|---|---|---|
| | *Pamungkas, Basile & Patti (2021a)* ♣ | En, Fr, De, Id, It, Pt and Es | Google Translate API to translate all datasets into En = input to BERT | 2021 |
| Ensemble learning (EL) | *Cohen et al. (2023)* ◇ | En, De, Fr, Es and No | Based DeBERTa: Simple averaging, weighted averaging based on AUC, and LightGBM using predictions as input. | 2023 |
| | *Ahn et al. (2020a)* ♣ | En, El, Da, Ar and Tr | Majority Voting based mBERT | 2020 |
| | *Bigoulaeva, Hangya & Fraser (2021)* ♣ | En and De | Based Bilingual word embeddings: FastText then MUSE | 2021 |
| | *Bigoulaeva et al. (2022)* ♣ | En and De | Based mBERT, CNN and LSTM (Cross-Lingual Word Embeddings) | 2022 |
| | *Bigoulaeva et al. (2023)* ♣ | En and Es | Based mBERT | 2023 |
| Meta learning (Meta-L) | *Vadakkekara Suresh, Chakravarthi & McCrae (2022)* ♣ | Ta-English and Malayalam-English code-mixed | MAML and Proto-MAML, based XLM-R | 2021 |
| | *Mozafari, Farahbakhsh & Crespi (2022)* ♣ | Hate speech: En, Ar, Es, De, Id, It, Pt, Fr and Offensive lang. Ar, Da, En, El, Fa and Tr | MAML and Proto-MAML, based XLM-R | 2022 |
| | *Awal et al. (2024)* ♣ | En, Es, Ar, Da, El, Tr, Hi, De, It | HateMAML: domain-adaptive MAML based mBERT and XLM-R | 2023 |
| Multitask-L—joint training | *Chiril et al. (2019)* ◇ | Fr and En | Based Bi-LSTM (Glove bilingual word embeddings) | 2019 |
| | *Pamungkas & Patti (2019)* ♣ | En, It, Es, and De | Based MUSE | 2019 |
| | *Pamungkas, Basile & Patti (2021a)* ♣ | En, Fr, De, Id, It, Pt and Es | Based MUSE, LASER, mBERT | 2021 |
| Multitask-L—auxiliary task | *Riabi, Montariol & Seddah (2022)* ♣ | En, It and Es | Based XLM-R, XLM-T | 2022 |
| | *Montariol, Riabi & Seddah (2022)* ♣ | En, It and Es | Based mBERT, XLM-R, XLM-T | 2022 |
| UL—GAE | *De la Peña Sarracén & Rosso (2022)* ◇ | En, De, Ru, Tr, Hr and Sq | Based mBERT, XLM-R (TFIDF) | 2022 |
| UL—adversarial | *Shi et al. (2022)* ♣ | En, Da, Ar, El and Tr | Based mBERT | 2022 |

**Notes:**
Only one representative study of each approach is cited in the table due to space limitation.
LR, logistic regression; TL, transfer learning; MT, machine translation; EL, ensemble learning; Meta-L, meta learning; Multitask-L, multitask learning; UL, Unsupervised learning; GAE, graph auto-encoders. Languages abbreviations are based on ISO 639 language codes list.
◇/♣: ◇ refers to Multilingual methods & ♣ refers to cross-lingual methods. Feature extraction methods are put between ().

more effectiveness in this field, especially by utilizing pre-trained language models. Traditionally, the most widely used machine learning techniques included Naive Bayes, k-nearest neighbors, decision trees, random forests, and support vector machines. However, in recent years deep neural networks have almost completely substituted or at least surpassed these methods, particularly in NLP and in sentiment analysis (*Otter, Medina & Kalita, 2018*). But traditional machine learning models might still be considered potential solutions to this issue because they are effective at identifying offensive language. On this scope, a multilingual hate speech and abusive language detection system was developed by *Vashistha & Zubiaga (2021)*, and trained on a significant textual dataset of hate speech in

English and Hindi. They demonstrated an online retraining capability for the system to identify new varieties of hate speech or linguistic patterns using LR. Moreover, cross-lingual (zero-shot and few-shot learning) experiments were executed by *Aluru et al. (2020)* on nine different languages. They analyzed different combinations of vector representations and machine learning algorithms, including MUSE and LASER embeddings. As a result, LASER and an LR model proved to be the most effective combined model. Adding to that, *Bigoulaeva, Hangya & Fraser (2021)* utilized a support vector machine (SVM), as a baseline classifier, along with bilingual word embeddings (BWE) to detect hate speech in English and German.

### Deep neural networks (DNN)

Deep neural networks have been widely used for multilingual offensive language recognition because of their ability to acquire complex representations of text across different languages. There was an extensive use of convolutional neural networks (CNNs) (*Elouali, Elberrichi & Elouali, 2020*; *Bigoulaeva et al., 2023*, *2022*; *Bigoulaeva, Hangya & Fraser, 2021*), CNN-GRU (gated recurrent unit) (*Deshpande, Farris & Kumar, 2022*; *Aluru et al., 2020*) and recurrent neural networks (RNNs) like long short-term memory (LSTM), where *Vashistha & Zubiaga (2021)* used CNN-LSTM model, and *Pamungkas, Basile & Patti (2021a)* utilized LSTM along with MUSE and mBERT. Adding to that, *Bigoulaeva et al. (2022)*, and *Vitiugin, Senarath & Purohit (2021)* used LSTM for word embeddings. As for bidirectional LSTM (BiLSTM), it was implemented by *Chiril et al. (2019)*, *Bigoulaeva et al. (2023)* and *Bigoulaeva, Hangya & Fraser (2021)*. Moreover, *Jiang & Zubiaga (2021)* proposed a hate speech detection model called CCNL-Ex that includes additional hate-related semantic features. The model uses a cross-lingual capsule network learning approach (CCNL) with two parallel architectures for source and target languages. They used BiLSTM to extract contextual features, and Capsule Network to capture hierarchically positional relationships.

### Transfer learning (TL)

Transfer learning has emerged as an effective approach for identifying multilingual hate speech because it enables systems to use the information obtained from data in the source domain to perform better on data in other domains. As a result, many studies employed Pre-trained multilingual word embeddings like FastText (*Bigoulaeva, Hangya & Fraser, 2021*), MUSE (*Pamungkas & Patti, 2019*; *Deshpande, Farris & Kumar, 2022*; *Aluru et al., 2020*; *Bigoulaeva, Hangya & Fraser, 2021*), or LASER (*Deshpande, Farris & Kumar, 2022*, *Aluru et al., 2020*, *Pelicon et al., 2021a*), and *Vitiugin, Senarath & Purohit (2021)*. Moreover, most of the research studies has focused on the use of pre-trained language models LLMs (basically as classifiers): BERT (*Vashistha & Zubiaga, 2021*, *zahra El-Alami, Ouatik El Alaoui & En Nahnahi, 2022*; *Zia et al., 2022*; *Pamungkas, Basile & Patti, 2021a*), AraBERT (for Arabic data) (*zahra El-Alami, Ouatik El Alaoui & En Nahnahi, 2022*), CseBERT (for English, Croatian and Slovenian data) (*Pelicon et al., 2021b*), as well as multilingual BERT models: (*Shi et al., 2022*; *Bhatia et al., 2021*; *Deshpande, Farris & Kumar, 2022*; *Aluru et al., 2020*; *zahra El-Alami, Ouatik El Alaoui & En Nahnahi, 2022*;

*De la Peña Sarracén & Rosso, 2022*; *Tita & Zubiaga, 2021*; *Eronen et al., 2022*; *Ranasinghe & Zampieri, 2021a*; *Ghadery & Moens, 2020*; *Pelicon et al., 2021b*; *Awal et al., 2024*; *Montariol, Riabi & Seddah, 2022*; *Ahn et al., 2020a*; *Bigoulaeva et al., 2022*, *2023*; *Pamungkas, Basile & Patti, 2021a*; *Pelicon et al., 2021a*), DistilmBERT model (*Vitiugin, Senarath & Purohit, 2021*), and RoBERTa (*Zia et al., 2022*).

On the other hand, cross-lingual language models like XLM were also widely employed, where we found implementation of XLM-RoBERTa (XLM-R) (*Roy et al., 2021a*; *Bhatia et al., 2021*; *Wang et al., 2020*; *De la Peña Sarracén & Rosso, 2022*; *Zia et al., 2022*; *Tita & Zubiaga, 2021*; *Ranasinghe & Zampieri, 2021b*; *Dadu & Pant, 2020*; *Eronen et al., 2022*; *Ranasinghe & Zampieri, 2021a*, *2020*; *Mozafari, Farahbakhsh & Crespi, 2022*; *Barbieri, Espinosa Anke & Camacho-Collados, 2022*; *Awal et al., 2024*; *Stappen, Brunn & Schuller, 2020*), and both XLM-R and XLM-T (*Montariol, Riabi & Seddah, 2022*, *Riabi, Montariol & Seddah, 2022*). These approaches have all been shown to improve performance on tasks involving multilingual/cross-lingual hate speech detection because they are more likely able to capture semantic and syntactic features across languages thanks to their pre-training on multilingual large volumes of texts. Therefore, transfer learning is expected to play an even greater part in improving the accuracy of multilingual hate speech detection algorithms.

### Machine translation (MT)

Using machine translation enables multilingual classification with monolingual models, where different languages are translated into the training language. Moreover, machine translation can be used as data augmentation to improve model performance. In this domain, *Jiang & Zubiaga (2021)* proposed a capsule network for cross-lingual hate speech detection. The network relies on source language and its translated counterpart in target language. *Aluru et al. (2020)* employed machine translation method for cross-lingual hate speech detection and compared the performance of LASER embedding and mBERT on datasets in 9 different languages. They found that simply adopting the machine translation method has comparative performance with multilingual models. *Pamungkas, Basile & Patti (2021a)* proposed a joint-learning architecture utilizing multilingual language representations, and evaluated several competitive baseline systems including using machine translation to augment training data. The authors further investigated the impact of integrating a multilingual hate lexicon as an external source of knowledge into their joint-learning models. They found that a simple model relying on automatic machine translation and an English BERT pre-trained model achieved competitive results in their tasks. *Ibrohim & Budi (2019b)* discussed the challenges of identifying hate speech in a multilingual setting and presented a comparison between two methods for multilingual text classification, translated and non-translated. The authors experimented with support vector machine, Naive Bayes, and Random Forest Decision Tree classifiers with word n-grams and char n-grams as feature extraction. The experiment results suggested that the non-translated method performs better, but it is more costly due to data collection and annotation. On the other hand, the translated method without language identification gives poor results. To address this issue, the authors proposed combining the translated

method with monolingual hate speech identification, which improved multilingual hate speech identification performance.

### Ensemble learning (EL)

In multilingual and cross-lingual settings, ensemble learning has shown promise as an approach for increasing the performance of offensive language detection systems. Recent studies have shown that researchers used a variety of methods to use ensemble learning in this domain. For instance, *Bigoulaeva et al. (2022)* used a bootstrapping ensemble of several models for unlabeled German datasets and then fine-tuned English-trained models using this bootstrapped data. Also, *Bigoulaeva, Hangya & Fraser (2021)* built a transferred system that used an ensemble-based approach to train on unlabeled data and included newly labeled data to improve performance on the target language. Adding to that, *Bigoulaeva et al. (2023)* provided a method for bootstrap labels using a variety of model structures and including unlabeled targeted language data for further advancements. Moreover, *Ahn et al. (2020a)* employed an ensembling procedure on multiple mBERT models to adjust hyperparameters (using the Translation Embedding Distance metric) and they improve the performance of both cross-lingual transfer and semi-supervised annotation labels. The model's performance was improved compared to the baselines (which were trained only on manually annotated data) after using the semi-supervised dataset. Finally, *Cohen et al. (2023)* utilized DeBERTa-based ensemble learning method, including both back-translation and GPT-3 augmentation.

### Meta learning (Meta-L)

Meta-learning, also known as "learning to learn" is a burgeoning field of machine learning that is concerned with developing algorithms that enable an agent to learn how to learn. The objective of meta-learning is to design models that can rapidly adapt to new tasks with limited training data by leveraging prior experience. Meta-learning has shown significant promise in the field of NLP, where it can be used to improve the performance of language models by leveraging knowledge gained from solving one task to improve performance on another related task, which is particularly useful in scenarios where there is limited labeled data available for a given task. Current meta-learning methods include optimization-based methods and metric-based methods. Optimization-based methods aim to learn better model parameter initialization (*Finn, Abbeel & Levine, 2017*), model architecture (*Zoph & Le, 2016*), or more efficient optimization strategy (*Andrychowicz et al., 2016*). Metric-based methods aim to learn better distance metrics (*Vinyals et al., 2016*) or representations (*Snell, Swersky & Zemel, 2017*) to enable more efficient data contrastiveness in the metric space.

Compared to the more general NLP tasks, there are much fewer works focusing on the use of meta-learning in hate speech detection, especially in the cross-lingual setting. *Vadakkekara Suresh, Chakravarthi & McCrae (2022)* proposed a two-step strategy using meta-learning algorithms to identify offensive text in Tamil-English and Malayalam-English code-mixed texts. The authors introduced a weighted data sampling approach to enable better convergence in the meta-training phase compared to conventional methods. Their experimental results demonstrated that the meta-learning approach improves the performance of models significantly in low-resource (few-shot learning) tasks.

*Mozafari, Farahbakhsh & Crespi (2022)* proposed a meta-learning-based approach to detect hate speech and offensive language in low-resource languages with limited labeled data. The methodology leverages two meta-learning models, MAML and Proto-MAML, to perform cross-lingual few-shot detection. The authors curated two diverse collections of publicly available datasets and compared the performance of their approach with transfer-learning-based models. The authors demonstrated by experiments that meta-learning-based models, particularly Proto-MAML, outperform transfer-learning-based models in most cases, and proposed that the meta-learning approach shows promise in identifying hateful or offensive content in low-resource languages with only a few labeled data points. *Awal et al. (2024)* proposed HateMAML to detect hate speech in low-resource languages. They proposed a self-supervision strategy to adapt the model to unseen target languages and evaluated the framework on five datasets across eight low-resource languages, which showed that the proposed meta-learning method outperformed state-of-the-art baselines in cross-domain multilingual transfer settings.

### Multitask learning (Multitask-L)

Using recent developments in NLP, multitask learning has come to be a promising strategy for enhancing multilingual and cross-lingual offensive language identification. This approach involves simultaneous training of models on multiple tasks in order to enhance their performance through the features sharing among the tasks data. Due to its capacity to enhance model performance by utilizing the knowledge gained from one or several tasks to improve the performance of another task, multitask learning has recently drawn more attention. Hate speech and offensive language detection models can benefit from this approach by training them on numerous related tasks such as sentiment analysis, dependency parsing, and named entity recognition, among others, allowing them to better detect offensive content. By training models on multiple tasks in multiple languages, multitask learning can enable these systems to better understand the nuances of different languages and improve their performance in multilingual offensive language (*Chen, Zhang & Yang, 2021*).

In this context, *Chiril et al. (2019)* used joint training on both English and 30% of the French dataset, then they tested the trained model on the rest of French *corpus*. In addition, *Pamungkas & Patti (2019)* used a domain-independent lexicon called "HurtLex" (*Bassignana, Basile & Patti, 2018*) in cross-domain and cross-language methods within a joint learning approach, which leads to improving the performance in detecting abusive content. Moreover, *Montariol, Riabi & Seddah (2022)* studied used multilingual model pre-trained models (XLM-R and XLM-T) within a multitask architecture. They analyzed the impact of auxiliary tasks, like named entity recognition (NER), part of speech (POS) tagging, dependency parsing, and sentiment analysis, and found that some hate speech labels were more susceptible to cross-lingual transfer learning. Furthermore, *Riabi, Montariol & Seddah (2022)* used XLM-R and XLM-t based multitask learning models, as well as the MACHAMP strategy to fine-tune on various auxiliary tasks. Lastly, *Pamungkas, Basile & Patti (2021a)* proposed two joint-learning approaches using diverse multilingual

language features to transfer knowledge between pairs of languages. According to their study, their models performed the best across the board.

### Unsupervised learning (UL)

The issue of labeled data insufficiency and data annotation can be handled by using methods of unsupervised learning that do not rely on labeled training data to detect multilingual hate speech and offensive language. In fact, as indicated in *De la Peña Sarracén & Rosso (2022)*, the authors use a label-free approach by encoding texts as graph nodes using Graph Auto-Encoders (GAE). This method utilizes a combination of transformers (mBERT and XLM-R) with convolutional neural layers, to encode the texts. Moreover, *Shi et al. (2022)*, proposed an unsupervised model that employs cross-lingual mapping, sample generation, and transfer learning. Their model employs a novel training methodology that combines adversarial learning, transfer learning, and agreement regularization to detect offensive language in many low-resource languages. In multilingual and cross-lingual environments, each of the aforementioned methodologies suggest interesting new directions for unsupervised hate speech identification.

### Other technologies

Several research studies have explored various advanced methods for offensive language detection and classification in different languages (especially in low-resource languages). In fact, a Transformer-based architecture called TIF-DNN was created for code-mixed Hindi and English (*Biradar, Saumya & Chauhan, 2021*) using translation and transliteration techniques for hate speech detection. Furthermore, AraBERT and MarBERT based multi-task learning models were presented in *Aldjanabi et al. (2021)*, these models perform offensive and hate speech detection tasks in modern standard Arabic language and in several dialects of Arabic tweets. Moreover, using polarity and emotions datasets, another multi-task learning technique (*Plaza-Del-Arco et al., 2021*) proved its success in the detection of hate speech in Spanish tweets. Additionally, BERT models were pre-trained on Hindi and Marathi: tweetsHindTweetBERT and MahaTweetBERT, illustrating state-of-the-art performance for hate speech detection in *Gokhale et al. (2022)*.

Since 2023, there have been many new approaches presented in the field of multilingual hate speech detection, emphasizing the necessity of more learning and more use of the new technological advancements. In fact, *Ghosal & Jain (2023)* introduced a new unsupervised approach in Hindi and Bengali languages, incorporating detection of hateful content, classification of tweets, and preparation of code-switch data. Furthermore, *Goldzycher et al. (2023)* utilizes intermediate English data fine-tuning along with Natural Language Inference (NLI) in Arabic, Portuguese, Spanish, and Italian hate speech detection. Adding to that, many BERT-based data augmentation methods are successfully incorporated to generate more data in various languages as illustrates in *Takawane et al. (2023)* where researchers, here, managed to enhance these models' performance on Code-Mixed Hindi-English hate speech data. Moreover, *Kar & Debbarma (2023)* explored a system using hybrid diagonal gated recurrent neural networks (DGRNN) within an optimal feature extraction technique in multilingual code-mixed texts in English, Hindi, and German.

Also, *Das, Pandey & Mukherjee (2023)* emphasized both the strengths and limitations of the ChatGPT model in hate speech detection in eleven languages. Lastly, *Roychowdhury & Gupta (2023)* presented many data-efficient techniques like task reformulation and data augmentation in French, Spanish, Arabic, and Portuguese in hate speech detection.

## Analysis of the existing approaches

Table 2 provides an overview of the previous research studies, we collected during our research analysis. It illustrate the different approaches implemented in multilingual and cross-lingual offensive language detection, we can see how these approaches have evolved over the years as well as the complexity, and the diversity of the languages studied in the field.

Around 2020, several approaches like *Vashistha & Zubiaga (2021)* and *Aluru et al. (2020)* concentrated on word embeddings for feature representation and they managed to use small-sized datasets in different languages such as English, Hindi, and some code-mixed languages. There is a progression towards more complex deep learning architectures like CNN-LSTM in *Vashistha & Zubiaga (2021)*, and diverse language sets in *Aluru et al. (2020)*, that were using more low resource languages like Arabic, Indonesian, Italian, German, Portuguese, Polish, French, and Spanish. Later on, 2021 witnessed the use of more complex approaches in research in this domain, such as the use of bi-directional pre-trained transformers (like mBERT) and the use of XLM-R, which displays more direction toward implementing pre-trained language models (especially the ones trained on significant volumes of multilingual data). More recently, since 2022, we observe more use of several sophisticated approaches, such as multitask learning, meta-learning, and ensemble learning. In particular, ensemble learning and meta-learning are gaining more attention, combining the predictions of multiple models or building and training a meta-learner able to adapt to new tasks with a few training examples, which could be mainly practical to use in the low-resource language data. Starting with the feature extraction techniques, word embeddings were among the widely used techniques, especially with the frequent implementation of several transformers like BERT and XLM-R, along with the use of other methods such as character-level representation and FastText but less often.

English language was the mostly learned compared to other languages, along with some other European languages such as Spanish, German, and French. Other languages were very little studied such as Japanese (Ja) and Norwegian (No), which reveals a serious challenge in analyzing offensive language and hate speech in these languages along with other non accessible ones (languages used by small communities that don't have ready data to work on). Therefore, we urge the need to conduct more research in order to be able to create performant services and tools for the detection of such content in these low-resource languages.

As shown in Table 2, the period of publication considered for the existing studies, ranges from 2019 and till the time of this study (July 2023). This implies that researching in multilingual text classification task is still relatively new area of study, with a lot of ongoing research. Moreover, for the multilingual approaches, the analyses cover a wide scope of languages, such as English, Bengali, Arabic, Danish, Croatian, French and more. As for the

cross-lingual approaches, the focus was mostly on fewer languages, typically, with English being the most studied as a source language (used for training), then some other rich-resource languages like Spanish, Italian, German, among others. Which indicates that multilingual text classification still requires much research, especially in low-resource languages. Lastly, we discovered an increasing tendency toward employing pretrained large language models (LLMs) like BERT and XLM-R and their variations (mBERT, XLM-T, among others). These LLMs prove their ability to remarkably enhance the performance of multilingual hate speech detection tasks.

## DATASETS ON MULTILINGUAL HATE SPEECH DETECTION

Most of the classification models for offensive language detection rely on supervised learning, therefore, access to high-quality and well-labeled data is necessary for training effective models. However, preparing such data is a very difficult and challenging task due to the enormous volume of information generated on social media platforms and other online sources. The task of curating and annotating this vast amount of data is both time-consuming and costly. In addition, a lot of effort is required to ensure the work is accurate and reliable. Thus, the process of data preparation continues to be a crucial and challenging aspect of training effective models.

The methodology of collecting the datasets analyzed in this section included first, selecting a set of chosen English keywords including terms like offensive language, hate speech, aggressive, multilingual, and low-resource datasets. Then utilizing these keywords we conducted searches on Google Scholar. Additionally, relevant workshops and shared tasks websites were also explored, as well as Hate Speech Dataset catalogue hatespeechdata, that presented many of these shared tasks datasets. The search process took place between March and April 2023. In the initial search round, we did not include any time or other filters and only considered the most relevant papers. Then to have also more recent datasets analyzed we applied a time filter to have more focus on the publications from 2020 to 2023. After the collection step, we created a table including the publication year, number of citations, and language(s) of each paper. We used this table to obtain and report the general statistics regarding the languages and citations of the datasets. Then due to time constraints, for more detailed analysis, we gave priority to papers representing datasets with more than one language and languages that had less than five datasets dedicated to them. For the remaining papers, we used the top ones with the most citations to ensure the inclusion of influential works. Furthermore, we also considered, in case of availability, the links provided in the papers, which were mostly from their GitHub repositories (URL to the repos are included in this study). This more detailed analysis led to the creation of Table 3 which we will explain in more detail.

Many of the analyzed hate speech datasets relied on Twitter as their primary data source. One key reason can be the availability of Twitter's public API (Application Programming Interface). This API allows researchers to retrieve relevant tweets based on specific criteria and keywords, including those related to offensive content, events, and target groups. After Twitter, Facebook pages were the next prominent source of offensive

**Table 3 Summary of the available datasets for hate speech detection.**

| Ref. | Year | Language(s) | Main subject | Source | Size | Cit. | Av. |
|---|---|---|---|---|---|---|---|
| *Carvalho et al. (2022)* | 2022 | Pt | HS | Twitter | 63,450 | <10 | N |
| *Wang, Day & Wu (2022)* | 2022 | Zh | HS | LINE today | 47,844 | <10 | N |
| *Ollagnier et al. (2022)* | 2022 | Fr | Agr. | Aggressive multiparty chats collected through a role-playing game | 19 conversations | <10 | Y |
| *Beyhan et al. (2022)* | 2022 | Tr | HS | Twitter | IstanbulConv:1206 Refugee:1278 | <10 | Y |
| *Madhu et al. (2023)* | 2023 | Hi-En | HS and OL | Twitter | 7,088 | <10 | Y |
| *Mohapatra et al. (2021)* | 2021 | Or/Or-En | HS | Facebook | 5,000 | <10 | N |
| *Steinberger et al. (2017)* | 2017 | Cs, En, Fr, It, De | Flames | User-generated news article discussions | Cs:1812 De:1122, En:1007 Fr:487 It:649 | <10 | Y |
| *Fernquist et al. (2019)* | 2019 | Sv | HS | A Swedish discussion forum | 3,056 | <10 | N |
| *Rahman et al. (2021)* | 2021 | En | HS | Twitter | 9,667 | <50 | Y |
| *Zampieri et al. (2022)* | 2022 | Mr | OL | Twitter | MOLD2.0: 3611 SeMold: 8000 | <50 | Y |
| *Nascimento et al. (2019)* | 2019 | Pt-BR | OL | Twitter and Brazilian 55chan imageboard | 7,672 | <50 | Y |
| *Akhtar, Basile & Patti (2021)* | 2021 | En | HS, Agr., OL, and stereotype | Twitter | 4,480 | <50 | N |
| *Mubarak, Hassan & Chowdhury (2022)* | 2022 | Ar | HS and OL | Twitter | 12,698 | <50 | N |
| *Ombui, Muchemi & Wagacha (2019)* | 2019 | En, Sw, other East African languages | HS and OL | Twitter | 260 k | <50 | N |
| *Evkoski et al. (2022)* | 2022 | Sl | HS | Twitter | 12,961,136 | <50 | Y |
| *Satapara et al. (2021)* | 2021 | Hi-En | HS | Twitter | 7,088 | <50 | Y |
| *Luu, Nguyen & Nguyen (2021)* | 2021 | Vi | HS and OL | Facebook and YouTube | 33,400 | <50 | Y |
| *Fanton et al. (2021)* | 2021 | En | HS/CN | | 5000 HS/CN pairs | <50 | Y |
| *Ali et al. (2022)* | 2022 | Ur | HS and OL | Twitter | 10,526 | <50 | N |
| *Ljubešić, Erjavec & Fišer (2018)* | 2018 | Sl, Hr | Moderated news comments | The Slovene RTV MCC and Croatian 24sata news portals | 24,639,651 | <50 | Y(1), Y (2) |
| *Vu et al. (2020)* | 2020 | Vi | HS and OL | Facebook | 5,431 | <50 | Y |
| *Ptaszynski, Pieciukiewicz & Dybała (2019)* | 2019 | Pl | HS and cyberbullying | Twitter | 11,041 | <50 | Y |
| *Gaikwad et al. (2021)* | 2021 | Mr | OL | Twitter | MOLD 1.0: 2499 | <50 | Y |
| *Haddad, Mulki & Oueslati (2019)* | 2019 | Tunisian Ar | HS and abusive | Different social media platforms | 6,075 | <50 | Y |
| *Das et al. (2021)* | 2021 | Bn | HS | Facebook | 7,425 | <50 | N |
| *Rizwan, Shakeel & Karim (2020)* | 2020 | Roman Ur | HS | Twitter | 10,012 | <50 | Y |
| *Guest et al. (2021)* | 2021 | En | Misogyny | Reddit | 6,567 | <50 | Y |

| Ref. | Year | Language(s) | Main subject | Source | Size | Cit. | Av. |
|---|---|---|---|---|---|---|---|
| *Leite et al. (2020)* | 2020 | Pt-BR | Toxic speech | Twitter | 21 K | <50 | Y |
| *Ishmam & Sharmin (2019)* | 2019 | Bn | HS | Facebook | 5,126 | <50 | Y |
| *Moon, Cho & Lee (2020)* | 2020 | Ko | Toxic speech (HS and OL) | A popular domestic entertainment news aggregation platform | 9,381 | <100 | Y |
| *Mandl et al. (2021)* | 2021 | En, Hi, Mr | HS and OL | Twitter | En:3843 Mr:1874 Hi:4594 | <100 | Y |
| *de Pelle & Moreira (2017)* | 2016 | Pt-BR | OL | Brazilian Web (g1.globo.com) | OFFCOMBR-2: 1250, OFFCOMBR-3: 1033 | <100 | Y |
| *Fortuna et al. (2019)* | 2019 | Pt | HS | Twitter | 5,668 | <100 | Y |
| *Álvarez-Carmona et al. (2018)* | 2018 | Es-MX | Agr. | Twitter | 10,856 | <100 | Y |
| *Fiser, Erjavec & Ljubešić (2017)* | 2017 | Sl | Socially unacceptable discourse | Spletno Oko1 (Web Eye) hotline service | 13,000 | <100 | N |
| *Mossie & Wang (2020)* | 2020 | Am | HS and vulnerable community | Facebook | 491,424 | <100 | N |
| *Kumar et al. (2020)* | 2020 | Bn, Hi, En | Agr. | YouTube | Approx. 6,000 per lang. | <100 | Y |
| *Ibrohim & Budi (2018)* | 2018 | Id | HS | Twitter | 2,016 | <100 | Y |
| *Mulki et al. (2019)* | 2019 | Levantine Ar | HS and abusive | Twitter | 5,846 | <150 | N |
| *Coltekin (2020)* | 2020 | Tr | OL | Twitter | 36,232 | <150 | Y |
| *Mathur et al. (2018)* | 2018 | Hi-En | HS and OL | Twitter | 3,679 | <150 | N |
| *Pitenis, Zampieri & Ranasinghe (2020)* | 2020 | El | OL | Twitter | OGTD 1.0: 4779, OGTD 2.0: 10287 | <150 | Y |
| *Chung et al. (2019)* | 2019 | En, Fr, It | HS/CN | Generated by experts | 4,078 HS/CN pairs | <150 | Y |
| *Sigurbergsson & Derczynski (2019)* | 2019 | Da | HS and OL | Reddit and Facebook | 3,600 | <150 | Y |
| *Pavlopoulos, Malakasiotis & Androutsopoulos (2017)* | 2017 | El | User comment moderation | A Greek news portal (http://www.gazzetta.gr/) | Approx. 1.6 M | <150 | Y |
| *Ibrohim & Budi (2019a)* | 2019 | Id | HS and abusive | Twitter | 13,169 | <150 | Y |
| *Pereira-Kohatsu et al. (2019)* | 2019 | Es | HS | Twitter | 6,000 | <150 | Y |
| *Kumar et al. (2018b)* | 2018 | Hi-En | Agr. | Facebook and Twitter | Approx. 18 k Tweets and 21 k Facebook comments | <150 | N |
| *Alfina et al. (2017)* | 2017 | Id | HS | Twitter | 520 | <200 | Y |
| *Ousidhoum et al. (2019)* | 2019 | En, Fr, Ar | HS | Twitter | En:5647 Fr:4014 Ar:3353 | <200 | Y |
| *Bohra et al. (2018)* | 2018 | Hi-En | HS | Twitter | 4,575 | <200 | Y |
| *Sanguinetti et al. (2018)* | 2018 | It | HS | Twitter | 6,009 | <200 | Y |
| *Fersini, Rosso & Anzovino (2018)* | 2018 | Es, En | Misogyny | Twitter | En:3977 Es:4138 | <250 | Y |
| *Mandl et al. (2019)* | 2019 | En, Hi, De | HS and OL | Twitter and Facebook | En:5852, Hi:4665, De:3819 | <350 | Y |

(Continued)

| Table 3 (continued) | | | | | | | |
|---|---|---|---|---|---|---|---|
| Ref. | Year | Language(s) | Main subject | Source | Size | Cit. | Av. |
| *Zampieri et al. (2020)* | 2020 | Ar, Da, En, El, Tr | OL | Twitter, Facebook, Reddit, a local newspaper: Ekstra Bladet | En:1448861, Ar:1589, Da:384, El:2486, Tr:6131 | <400 | Y |
| *Del Vigna et al. (2017)* | 2017 | It | HS | Facebook | ? | <400 | N |
| *Basile et al. (2019)* | 2019 | Es, En | HS | Twitter | En:13000, Es:6600 | <750 | Y |

Note:
For the column names, Ref, reference; Cit., citation by May-2023, and Av., available. Language names have been shortened using the ISO 639-1 standardized nomenclature. Under the "Main Subject" column: HS, hate speech; OL, offensive language; CN, counter-narrative, and Agr., aggressiveness. In "Size" column: Approx., approximately. In "Av." column: Y, Yes and N, No. Links to the resources may not been shown in the hard copy.

language collected corpora. Offensive language datasets have also incorporated data from platforms like YouTube and Reddit, alongside various other sources and websites.

The collected datasets encompassed a variety of subjects and used different terms to describe the types of offensive content they gathered, highlighting the different aspects of negative language prevalent in online discourse. Many of the datasets specifically focused on hate speech, offensive language, and aggressiveness. Others explored misogyny, cyberbullying, abusive language, socially unacceptable discourse, moderated news comments, stereotypes, among others. Datasets focusing on offensive content have also made efforts to encompass a diverse range of populations, taking into account various characteristics such as race, gender, ethnicity, religion, and sexual orientation. Women, individuals of African ancestry, LGBTQ+ individuals, immigrants, and members of various religious organizations, including Hindus, Christians, Jews, and Muslims are some of the highly targeted groups. By including such diverse populations, these datasets aim to provide a comprehensive understanding of offensive language and its impact on different communities.

Offensive language datasets come with diverse labeling schemes, capturing the multifaceted nature of the content analyzed. Table 4 provides an overview of datasets that encompass various types of labeling schemes, including binary labels, intensity levels, variations in hateful speech categorization, and multiple labels for themes and target groups. Most of the datasets rely on binary labels like hate/non-hate, offensive/non-offensive, aggressive/non-aggressive, among others. A number of datasets also annotated the levels of hate showing how weak or strong the hate and offensiveness is significantly providing a more detailed understanding of the intensity of harmful speech. Some datasets exhibit variations in labeling strategies particularly in distinguishing between different forms of negative speech. These variations can include the presence of distinct labels for hate speech, offensive language, abusive speech, among others. To address the context of hate speech, some datasets introduce multiple labels that capture themes and target groups of hate speech or whether they are aimed at individuals or groups.

Table 3 presents a comprehensive and structured analysis of hate speech detection datasets analyzed in this study. The table provides essential information for each dataset, including the year of publication, languages covered, and the main subject, encompassing hate speech, offensive language, aggressiveness, and more. Additionally, the table includes details on the dataset source, indicating the platform from which the data was collected,

**Table 4 Type of available labels in the studied Datasets.**

| Dataset labeling schemes | Datasets |
|---|---|
| Only binary labels | *Zampieri et al. (2020)*, *Bohra et al. (2018)*, *Alfina et al. (2017)*, *Pitenis, Zampieri & Ranasinghe (2020)*, *Álvarez-Carmona et al. (2018)*, *Pereira-Kohatsu et al. (2019)*, *Pavlopoulos, Malakasiotis & Androutsopoulos (2017)*, *Ptaszynski, Pieciukiewicz & Dybała (2019)*, *de Pelle & Moreira (2017)*, *Evkoski et al. (2022)*, *Steinberger et al. (2017)*, *Rahman et al. (2021)*, *Nascimento et al. (2019)*, *Wang, Day & Wu (2022)*, *Yang, Jang & Cho (2022)*, *Ranasinghe et al. (2022)*, *Madhu et al. (2023)*, *Satapara et al. (2021)*, *Aliyu et al. (2022)*, *Gaikwad et al. (2021)* |
| Contains intensity levels | *Del Vigna et al. (2017)*, *Sanguinetti et al. (2018)*, *Ibrohim & Budi (2019a)*, *Kumar et al. (2020)* |
| Contains different categorizations of negative speech | *Sanguinetti et al. (2018)*, *Mandl et al. (2019)*, *Ousidhoum et al. (2019)*, *Ibrohim & Budi (2019a)*, *Mulki et al. (2019)*, *Basile et al. (2019)*, *Ibrohim & Budi (2018)*, *Fersini, Rosso & Anzovino, 2018*, *Mandl et al. (2021)*, *Haddad, Mulki & Oueslati (2019)*, *Moon, Cho & Lee (2020)*, *Mathur et al. (2018)*, *Vu et al. (2020)*, *Luu, Nguyen & Nguyen (2021)*, *Ombui, Muchemi & Wagacha (2019)*, *Das et al. (2021)*, *Mohapatra et al. (2021)*, *Beyhan et al. (2022)*, *Ali et al. (2022)*, *Fernquist et al. (2019)*, *Mubarak, Hassan & Chowdhury (2022)*, *Mazari & Kheddar (2023)*, *Akhtar, Basile & Patti (2021)*, *Rizwan, Shakeel & Karim (2020)* |
| Contains themes/Target groups | *Del Vigna et al. (2017)*, *Ibrohim & Budi (2019a)*, *Sigurbergsson & Derczynski (2019)*, *Coltekin (2020)*, *Basile et al. (2019)*, *Mossie & Wang (2020)*, *Fortuna et al. (2019)*, *Kumar et al. (2018b)*, *Fersini, Rosso & Anzovino (2018)*, *Ishmam & Sharmin (2019)*, *Das et al. (2021)*, *Beyhan et al. (2022)*, *Guest et al. (2021)*, *Carvalho et al. (2022)*, *Yadav et al. (2023)*, *Akram, Shahzad & Bashir (2023)*, *Zampieri et al. (2022)*, *Rizwan, Shakeel & Karim (2020)* |

such as Twitter or Facebook. Dataset size is also included, representing the number of samples within each dataset, while the citation count provides a measure of the usage and recognition of the datasets. Furthermore, the table indicates the availability of each dataset along with the link where they can be found, the publicly accessible ones are marked as "Y" with a hyperlink, and those that are not are marked as "N". This comprehensive overview serves as a valuable resource, offering researchers a consolidated reference for hate speech detection datasets, their attributes, and accessibility.

We constructed an informative figure to depict the availability of datasets for different languages and the corresponding citations, enabling us to assess the corpora distribution and utilization patterns across various languages. Figure 2 showcases the languages for which datasets are available, alongside the number of citations received by each dataset's paper. This visual representation offers valuable insights into the high and low-resource languages, emphasizing the significance of both categories in research endeavors. It allows us to identify languages that receive substantial attention and recognition, regardless of their resource availability. The figure underscores the importance of supporting research efforts for low-resource languages, as their impact and usage transcend their limited resources. The horizontal bar chart displays the languages on the y-axis, while the x-axis represents the number of papers available for each language. Each bar's color is determined by the corresponding number of citations received (by the time of this study: May 2023), with a color bar provided to indicate the intensity of citation impact. Please note that English has been intentionally omitted from the chart to maintain visual clarity. With 47 papers and approximately 11,000 citations, English's strong presence would have overshadowed the statistics of other languages, limiting the informative value of the figure. During the analysis of the collected datasets, it was observed that English, Italian, and Arabic were the most prevalent languages, with a relatively high number of dedicated

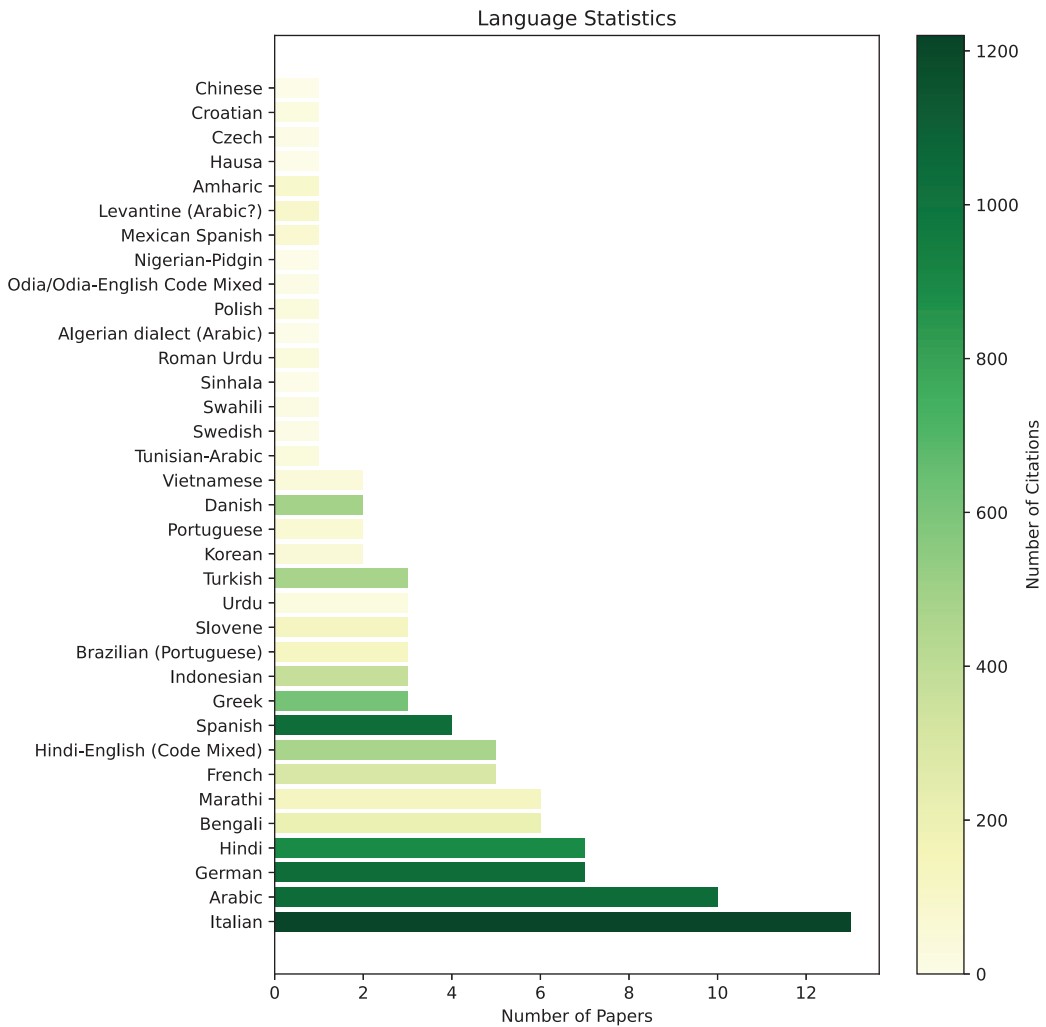

**Figure 2 Datasets distribution based on languages.** Number of papers studies in this survey and their impact (based on citations @May 2023). Note: English (47 papers with 11 k citations) is excluded from the figure to have a better visualization in the distribution.

datasets available for each. Conversely, several languages were represented by only a single dataset. Notably, some languages such as Spanish, Greek, Indonesian, Turkish, and Danish, despite having more limited dedicated datasets, received a considerable number of citations. This observation points to a noteworthy level of interest and utilization within the research community for these languages. It emphasizes the importance and necessity of addressing the needs of low-resource languages.

## RESOURCES FOR MULTILINGUAL HATE SPEECH DETECTION

In this section, we aim to explore the existing resources on the multilingual offensive language field, we divide this section into two major parts: available collaborative projects, and related products (*e.g.* open community challenges, source codes, and APIs). We are strengthening these resources' influence by bringing them to the awareness of a wide

audience, including developers, academics, students, among others, who can employ these tools to work on offensive language detection in multiple languages.

## Collaborative international projects

A summary of the projects we are mentioning in this section is presented briefly in Table 5.

**European project:** Several European projects have been undertaken to address the detection and mitigation of online hate speech. The *DARE* project, https://cordis.europa.eu/project/id/725349 (2017–2021), funded under the EU Horizon 2020 Framework Programme, aims to develop technologies and methodologies for combating hate speech, including radicalization and extremist content. It involves partners from 13 countries, including Belgium, Croatia, Germany, Greece, France, Malta, Poland, Russia, Turkey, Tunisia, The Netherlands, and the UK. Another project, *Hatemeter* (2018–2020), focuses on monitoring, analyzing, and tackling anti-Muslim hatred online at the EU level. It takes a multidimensional approach to identify red flags of hate speech, understand patterns of Islamophobia, develop tactical and strategic responses, and produce counter-narratives. The project partners include Italy, France, and the UK. In the realm of cybersecurity, the *PANACEA* (2019–2022) project aims to improve cybersecurity and privacy/data protection in hospital and health infrastructures. It provides toolkits to enhance security and data protection for various stakeholders in the healthcare sector, including hospitals, software/system developers, medical device manufacturers, and digital service providers. The project involves partners from Italy, UK, Greece, France, Belgium, Netherlands, Germany, and Ireland. Moreover, the *sCAN* project (2018–2020), coordinated by LICRA (International League against Racism and Antisemitism), focuses on gathering expertise, tools, methodology, and knowledge to identify, analyze, report, and counteract online hate speech. It involves partners from France, Germany, Italy, Belgium, Czech Republic, Austria, Slovenia, Croatia, and Latvia.

While explicit information about language support may not be mentioned for some of the mentioned projects above, it is reasonable to assume that their deliverables and solutions would likely focus on the languages spoken in the countries of their partner organizations. Given the diverse range of partner countries involved in these projects, it is possible that they would consider the languages relevant to those countries. This would imply that their solutions and products could potentially support languages beyond English, depending on the specific project's objectives and target regions.

Other projects are explicitly mentioning their focus on various languages. Among them, we found Detect Then Act (DTCT) (2019–2021), a European collaboration that aims to monitor and tackle online hate speech. It utilizes Explainable AI to assist users in deflating toxic discussions. The project reports illegal hate speech cases in accordance with the EU's Code of Conduct and local legislation. Moreover, it provides master training for hate speech detection in multiple languages, including English, French, Dutch, German, and Hungarian. The project partners involved are from Belgium, Germany, the UK, and the Netherlands. The project Stand By Me (2020) was created to moderate online violence against women in Europe. The project aims to help addressing this issue by enhancing individuals' awareness and capability to recognize such content. The project utilizes a

**Table 5 Collaborative international projects on hate speech.**

| Project | Partners | Year |
|---|---|---|
| DARE: Dialogue About Radicalisation and Equality | Belgium, Croatia, France, Germany, Greece, Malta, Norway, Poland, Russian Federation, The Netherlands, Tunisia, Turkey and the UK | 2017–2020 |
| MANDOLA: Monitoring and Detecting OnLine Hate Speech | Greece, Ireland, France, Spain, Bulgaria and Cyprus | 2017 |
| PRO2HATERS: PROactive PROfiling of HATE speech spreadeRs | Germany | 2017 |
| Hatemeter: hate speech tool for monitoring, analyzing and tackling anti-Muslim hatred online | Italy, France and the UK | 2018–2020 |
| sCAN: specialised cyber-activists network | France, Germany, Italy, Belgium, Czech Republic, Austria, Slovenia, Croatia and Latvia | 2018–2020 |
| PANACEA: Protection and privAcy of hospital and health iNfrastructures with smArt Cyber sEcurity and cyber threat toolkit for dAta and people | Italy, UK, Greece, France, Belgium, Netherlands, Germany and Ireland | 2019–2022 |
| DTCT: Detect Then Act | Belgium, Germany, the UK and the Netherlands | 2019–2021 |
| Stand By Me | Italy, Poland and Hungary | 2020 |
| EOOH: The European Observatory of Online Hate | Belgium, Slovakia and the Netherlands | 2021 |
| Identrics | Bulgaria | 2023 |
| OHI: Online Hate Index | USA | Released in 2018 |
| ProPublica's Documenting Hate project | USA | Started in 2017 |

diverse approach, including a combined learning program and educational resources. In addition, The European Observatory of Online Hate (EOOH) project (2021) was released by Textgain (in the lead), as a Multi-platform for monitoring hate speech in more than 20 social media platforms and covering 24 different languages. This project has made significant progress in understanding the complexities of online hate speech. This incorporates comprehending its various forms, the relationships among corresponding users, and the strategies involving disinformation. Lastly, the *MANDOLA* project (2017) aims to improve our understanding of online hate speech prevalence and empower ordinary citizens to report it. It utilizes big-data approaches to monitor this content, provide policymakers with actionable information, and transfer best practices among Member States. The project partners include Greece, Ireland, France, Spain, Bulgaria, and Cyprus. While the specific languages supported by the project may not be explicitly mentioned in the provided information, the project's objectives display that it seeks to monitor and analyze hate-related speech across multiple languages, including languages other than English.

On the other hand, several "industrial projects" aim to tackle the issue of hate speech and polarization in society. One such project is PRO2HATERS: PROactive PROfiling of HATE speech spreadeRs (2017) by Symanto, (in Germany). PRO2HATERS focuses on addressing hate speech and polarization, with a particular emphasis on languages beyond English, such as German and Spanish, and their dialects. The project proposes language resources, network analysis, methods, and tools as key components to combat hate speech. It envisions various application scenarios, including cyber-security, where government

agencies can detect and counter hate speech, and social media companies can automate hate speech detection. Another project in this domain is Identrics (2023, Bulgaria). Identrics offers a cutting-edge Hate Speech Detection service that helps eliminate hate speech in the comments sections of websites. Leveraging machine learning models, this project continuously learns to identify and flag hate speech, providing alerts to potential occurrences. By utilizing Identrics' service, platforms can foster meaningful conversations without the concern of offensive or abusive language spreading throughout their community.

**Non-European projects:** Fighting hate speech is undoubtedly an international effort that cuts across national boundaries. The multifaceted projects use a wide range of tactics, including constructing machine learning models, crafting legislation, starting public education programs, and starting awareness campaigns. Despite the complexity of the problem, these international initiatives show a shared dedication to creating safer and more inclusive online and physical settings. The Online Hate Index (OHI), a tool employing machine learning to identify and quantify hate speech targeting marginalized groups on digital platforms in English, it was created by the Anti-Defamation League ADLs's Center for Technology and Society (CTS). The program is made to identify linguistic trends and continuously advance in antisemitic content detection, giving an objective way to gauge the incidence of hate speech and assess the success of digital businesses' anti-hate measures. This project is made by the USA and released in 2018 (the project was developed domestically but has international implications to be used worldwide). Moreover, ProPublica's Documenting Hate project, a well-known American endeavor that was started in 2017, aimed to compile an extensive database of hate crimes committed throughout the nation. The project teamed up with newsrooms, educational institutions, and independent journalists to assist in reporting and documenting instances of bias and hatred in response to the dearth of accurate statistics on hate crimes.

## Available products

We examine, in this part, various facets of the resources available in hate speech detection, more specifically in multilingual hate speech detection. We'll start by highlighting the community challenges and competitions. We will next move on to talking about the accessible open-source codes. These represent concrete instruments that are open to learning and discovering the developed solutions. In order to wrap off this analysis, we will look at the APIs, including multilingual APIs.

**Community Challenges & Datasets Provided:** Detecting multilingual hate speech and offensive language is a paramount challenge, especially for social media platforms where a myriad of cultures and languages interact daily. This complexity arises due to the nuanced, context-specific, and often indirect nature of hate speech and offensive language. Over the years, several competitions and hackathons have been aimed at addressing this issue. In 2018, the TRAC-1: Aggression Identification (*Kumar et al., 2018a*), the first Workshop on Trolling, Aggression, and Cyberbullying, honed in on identifying aggression in social media posts in English and Hindi, both in Roman and Devanagari scripts. Then, 2019 saw many important challenges, such as: the SemEval-2019 Task 5: Multilingual Detection of

Hate Speech Against Immigrants and Women in Twitter (*Basile et al., 2019*). It centered around hate speech detection in a multilingual setting, focusing on English and Spanish tweets. In 2020, several key events occurred. Kaggle's Jigsaw Multilingual Toxic Comment Classification competition encouraged participants to build models that identify rudeness, disrespect, or any conversation-derailing toxicity in multilingual online discussions using English-only training data. The same year, the Hate Speech Detection (HASOC) Competition, hosted by FIRE (Forum for Information Retrieval Evaluation), focused on identifying hate speech and offensive content in English, German, and Hindi. Additionally, the TRAC 2020, the second Workshop on Trolling, Aggression, and Cyberbullying, provided two shared tasks, one on Aggression Identification and another on Misogynistic Aggression Identification in Bangla (in both Roman and Bangla script), Hindi (in both Roman and Devanagari script) and English. Adding to that, in 2021, two significant challenges emerged: the Kaggle IIIT-D Multilingual Abusive Comment Identification focused on identifying abusive comments across various Indic languages, and the PAN shared task of Profiling Hate Speech Spreaders on Twitter 2021 involved profiling hate speech spreaders on Twitter in English and Spanish. Apart from these targeted events, numerous other hackathons and competitions contribute indirectly to the field of multilingual hate speech detection. Events centered around cross-lingual or multilingual text classification, sentiment analysis, or broad NLP problems offer valuable platforms for devising innovative solutions to detect hate speech and offensive language across languages.

**Datasets provided of the community challenges:** Overall, the above-mentioned challenges are presented in Table 6, they were instrumental in providing a broad spectrum of datasets that cater to different languages and aspects of hate speech detection. They are sources of rich and diverse information, accessible to researchers worldwide. By compiling and making these datasets available, they have fundamentally contributed to the field. These datasets can be accessed by registering or filling out the appropriate forms. One prominent resource comes from the SemEval 2019 Task 5, a Shared Task focused on the Multilingual Detection of Hate. Furthermore, the IIT-D Multilingual Abusive Comment Identification challenge has provided a dataset unique in its capacity for massively multilingual abusive comment identification across a variety of Indic languages. Another dataset worth mentioning revolves around Profiling Hate Speech Spreaders on Twitter, and is available in both English and Spanish. Complementing this, the HASOC 2020 challenge provides a dataset for Hate Speech and Offensive Content Identification. Additionally, the dataset from the TRAC—2020, caters to Bangla, Hindi, and English.

**Available source codes:** Table 7 provides a general overview of different solutions that have been developed in multilingual hate speech and offensive language detection. The source codes presented here are across 2020 to 2023, dealing with a variety of languages. Recently, *Cohen et al. (2023)* offered a source code along with live demonstrations to execute it. This code especially helps to further study and implement ensemble models based on RoBERTa or DeBERTa. It also gives a practical tool for researchers studying back translation and GPT-3 data augmentation techniques. Also, *Deshpande, Farris & Kumar (2022)* provided the source code of a model for detecting hate speech across ten languages.

**Table 6 Summary of community challenges on multilingual hate speech detection.**

| Challenge name | Languages | Year |
|---|---|---|
| TRAC-1: aggression identification | English, Hindi (Roman and Devanagari scripts) | 2018 |
| SemEval-2019 Task 5: multilingual detection of hate speech against immigrants and women in Twitter | English, Spanish | 2019 |
| Kaggle's jigsaw multilingual toxic comment classification | Multilingual (trained on English data) | 2020 |
| Hate speech detection (HASOC) competition | English, German, Hindi | 2020 |
| TRAC 2020: trolling, aggression, and cyberbullying | English, Hindi (Roman and Devanagari scripts), Bangla (Roman and Bangla scripts) | 2020 |
| Kaggle IIIT-D multilingual abusive comment identification | Multiple Indic languages | 2021 |
| PAN shared task of profiling hate speech spreaders on Twitter | English, Spanish | 2021 |

**Table 7 GitHub repositories for multilingual source code projects of hate speech.**

| Name | Languages | Link | Study if available | Year |
|---|---|---|---|---|
| Enhancing social network hate detection using back translation and GPT-3 augmentations during training and test-time | En, Fr, De, Es and No | Code | *Cohen et al. (2023)* | 2023 |
| Highly generalizable models for multilingual hate speech detection | En, Ar, De, Id, It, Pt, Es, Fr, Tr, Da and Hi | Code | *Deshpande, Farris & Kumar (2022)* | 2022 |
| Multilingual-abuse-comment-detection | 17 Hi languages | Code | Non available | 2022 |
| HateCheck: functional tests for hate speech detection models | Ar, Nl, Fr, De, Hi, It, Zh, Pl, Pt and Es | Code | *Röttger et al. (2021)* | 2022 |
| Deep learning models for multilingual hate speech detection | Ar, En, De, Id, It, Pl, Pt, Es and Fr | Code | *Aluru et al. (2021)* | 2021 |
| EACL 2021 OffensEval in Dravidian languages | Kn, Ml and Ta | Code | *Jayanthi & Gupta (2021)* | 2021 |
| Leveraging multilingual transformers for hate speech detection | En, De and Hi | Code | *Roy et al. (2021b)* | 2021 |
| Offensive language detection from multilingual code-mixed text using transformers | Kn, Ml and Ta | Code | *Sharif, Hossain & Hoque (2021)* | 2021 |
| Multi-oli | Da, Ko and En | Code | The language-adversarial training pipeline inspired from *Keung, Lu & Bhardwaj (2019)* | 2021 |
| Indonesian text classification multilingual | En and Id | Code | *Putra & Purwarianti (2020)* | 2021 |
| Detoxify: toxic comment classification with Pytorch lightning and transformers | En, Fr, Es, It, Pt, Tr and Ru | Code | Non available | 2020 |
| NLPDove at SemEval-2020 Task 12: Improving offensive language detection with cross-lingual transfer | En, El, Da, Ar and Tr | Code | *Ahn et al. (2020b)* | 2020 |
| Multilingual fairness LREC | En, It, Pl, Pt, Es | Code | *Huang et al. (2020)* | 2020 |

Another GitHub project is the "Multilingual-Abuse-Comment-Detection", which focuses on identifying abusive comments in seventeen Indian languages using MuRIL-based models (BERT based Multilingual Representations for Indian Languages). Moreover, *Röttger et al. (2021)* evaluated the performance of hate speech detection models across ten languages, giving code to test across multilingual provided datasets. Adding to that, *Aluru et al. (2021)* worked across nine languages on sixteen datasets for the classification of hate speech data, presenting a source code to train and fine-tune several models, including

mBERT-based, Translation+BERT, CNN+GRU and LASER+LR. Besides, each of *Aluru et al. (2021)* and *Sharif, Hossain & Hoque (2021)* gave solutions for offensive language detection in three different Dravidian languages. In 2020, the "Detoxify" project by *Hanu & Unitary Team (2020)*, established on three Jigsaw challenges, studied multilingual toxic comment classification across seventeen languages using XLM-R based models, as well as *Ahn et al. (2020b)* which deals with offensive language detection in five languages, within Semeval 2020 task, being among the first ten places in each of Greek, Danish, and Turkish languages datasets. Overall, these source codes describe the recent research actions toward producing more practical and effective solutions for multilingual hate speech and offensive language detection. They underscore the increasing direction toward low-resource languages. Even though some of these codes didn't have research study associated, they present detailed descriptions of their source codes in GitHub repositories.

**Available APIs:** The landscape of hate speech detection is rich with an array of tools that harness the power of AI to identify and counteract such harmful discourse. A comprehensive collection of tools and services designed to counteract and analyze hate speech is available online. Among the many prominent tools, illustrated in Table 8, we have the HateLab, an international center for studying hate speech founded by the Economic and Social Research Council (ESRC), the RapidAPI Hate Speech Detection that enables effective detection of offensive language, and iSpotHate, a freely available API dedicated to eradicating hate speech. Moreover, the Python library HateSonar (2020) offers simple and efficient hate speech detection without any need for user training, and Profanity-check (2019), another one, swiftly checks for profanity or offensive language in strings. Furthermore, StopPropagHate by INESC TEC, utilizes machine learning techniques to help news organizations automatically identify hate speech. *Cohere* offers a text moderation API that can efficiently filter out harmful or inappropriate content in real-time, while *Hive.ai* is a high-speed content moderation API with extensive training data (with results returned in under 200 ms). Additionally, MODERATION API can detect and hide a wide range of data entities, including sensitive information and inappropriate content. Similarly, Openai contributes to comprehend linguistic context, precisely identifying subtle instances of abusive language, including hate speech, cyberbullying, and content that promotes self-harm, as well as detecting probable instances of misinformation or disinformation.

**Multilingual APIs:** The relevance of multilingual tools in hate speech detection is paramount, as they aid in breaking language barriers to ensure the internet remains a safe space for all. Among these powerful tools, Sightengine stands out with its capability to detect hateful, sexual and toxic content across multiple languages, which include not just English, but also Chinese, French, Italian, Dutch, German, Portuguese, Swedish, Turkey, Filipino, and Spanish. Similarly, Spectrum Labs' Guardian elevates the standard of multilingual content moderation. Unlike conventional tools that largely depend on keyword-based filters, Guardian employs true Natural Language Understanding (NLU) AI. This advanced technique allows the system to detect harmful behaviors, such as bullying, hate speech, spam, extremism, among others, across languages including Arabic, French, Hindi, Korean, among many others. Microsoft Azure is an API developed within

Table 8 Summary of APIs and their language capabilities in hate speech detection.

| API name | Description | Multilingual support |
| --- | --- | --- |
| HateLab | International center for studying hate speech | – |
| RapidAPI hate speech detection | Detection of offensive language | – |
| iSpotHate | Detection and elimination of hate speech | – |
| HateSonar | Python library—hate speech detection | – |
| Profanity-check | Python library—profanity detection | – |
| StopPropagHate | Hate speech detection and prediction of a news potential to provoke such comments. | – |
| Cohere | Filter out harmful or inappropriate content | – |
| Hive.ai | High-speed content moderation API | – |
| MODERATION API | Detect and hide sensitive and inappropriate content | – |
| Openai | Detection of abusive language, hate speech, and misinformation. | – |
| Sightengine | Detection of hateful, sexual and toxic content | English, Chinese, Dutch, French, German, Portuguese, Italian, Swedish, Spanish, Tagalog/Filipino and Turkish. |
| Spectrum labs' guardian | Employs natural language understanding AI to detect harmful behaviors. | Arabic, French, Hindi, Korean, and many others |
| Microsoft azure | Detection of profanity | Over 110 languages |
| Membrace | Filter out various types of hateful content | English, Spanish, German, French, Polish, Turkish, Dutch, Italian, Swedish, Arabic, Chinese, Portuguese, Japanese, and Russian. |
| Alibaba cloud's text moderation 2.0 | Content review, and custom configurations | Up to 20 languages |
| Huawei cloud | Content moderation API | Chinese |

Cognitive Services, it helps in detecting profanity in more than 110 languages. Also, Membrace can filter out various types of content, including spam, clickbait, offensiveness, among others. It covers multiple languages: English, German, French, Polish, Spanish, Turkish, Italian, Dutch, Swedish, Chinese, Arabic, Chinese, Russian and Japanese. Moreover, Alibaba Cloud's Text Moderation 2.0 API, which supports up to 20 languages, offers a potent suite of features. These include content review, and custom configurations. Adding to that, Huawei Cloud contributes to this language-inclusive trend with its content moderation API. Although it currently supports only Chinese, its presence underlines the importance of multilingual tools and the continuous strides being made towards expanding language support in the field of hate speech detection.

## CHALLENGES AND LIMITATIONS

In this section first, we review the main challenges that have been faced during the detection of offensive language in NLP, with a focus on the challenges in multilingual and cross-lingual corresponding tasks. Next, we examine limitations, and lastly, we propose some future directions.

## Challenges

### Technical challenges

**Lack of labeled data across languages:** One of the main issues lies in the scarcity of annotated data, or the non-accessible ones (non-public ones). Thus, getting this data proves to be a difficult and time-consuming task. The restricted availability of such data, and in certain languages acts as a significant obstacle. Building highly accurate and performant models requires often a large amount of annotated data, especially in the target languages. Nevertheless, the process of data annotation is still challenging; it is costly, time-consuming, and requires a lot of experts in the domain to do the job (*Kovács, Alonso & Saini, 2021*; *Röttger et al., 2022a*), especially with the granularity of this content (*Vidgen et al., 2019*). Moreover, the problem of imbalanced datasets still persists, usually making the offensive labeled data in all its categories a minor class. Therefore, traditional machine learning techniques often perform badly on these minority class samples, especially in binary datasets. Researchers have suggested a number of oversampling and undersampling strategies to solve this issue as in *Khairy, Mahmoud & Abd-El-Hafeez (2024)*.

Adding to that, even non-English datasets present a substantial challenge, as there are still limited annotated datasets in the domain. For example, a dataset could include tweets in Persian and Arabic while creating a dataset of hate speech in Urdu (*Ali et al., 2022*). As a result, taking these challenges into consideration, it becomes obvious that the creation of annotated datasets in multiple languages remains an important step for advancing approaches, especially in low-resource languages like Arabic (*Omar, Mahmoud & Abd-El-Hafeez, 2020*), among others.

**Cross-lingual transfer learning:** Applying knowledge learned from one language to another, is considered a difficult task. Despite the recent significant progress to build accurate pre-trained multilingual language models, their cross-lingual ability for offensive language detection remains limited. This limitation is evident when working on swear words of specific cultures, which often vary among languages, and cannot even be easily translatable with the current machine translation tools. For instance, researchers have found important linguistic problems when employing Google Translate in their models, more specifically, they identified errors (*Pamungkas & Patti, 2019*). Another crucial problem is the unstable performance of some approaches across distinct target languages. In fact, *Glavas, Karan & Vulić (2020)* indicates that rich-resource languages manage to give better results compared to low-resource ones.

**Language and topic inequality:** The dominance of the English language in the current datasets has led to another significant challenge, such as anglophone bias outcomes in non-English data. This issue affects prominent companies such as Facebook, whose capacities were limited in 2020 to detect hate speech in Spanish, and Mandarin (*Aluru et al., 2020*). Adding to that, datasets in other languages are not only insufficient, but also tend to be small-sized, restricting the performance of offensive content detection in these languages (*Aluru et al., 2020*) which explains the restricted number of studies in these low-resource languages, like the Arabic language (*Khairy, Mahmoud & Abd-El-Hafeez, 2021*). Another crucial factor is the dynamics of language and topic, as some datasets just cover one topic

(misogyny, racism, among others) in many languages (*Arango, Pérez & Poblete, 2019*), which adds more complexity to the generalisability of this detection task in multilingualism.

**Bias:** Bias can appear during data collection, labeling, or training. It is one of the main problems that makes it hard to identify offensive language in multiple languages. A number of biases were found: racial bias, author bias, and subject bias. However, topic bias is still the most important one, as shown by some studies in this field (*Arango, Pérez & Poblete, 2019*). As one of the vital solutions to this issue, many studies have introduced several functional tests, such as *Röttger et al. (2022b)*, which have presented functional tests for hate speech detection models, introducing Multilingual HateCheck (MHC). Their work offered a various set of tests across ten different languages and aimed to improve the assessment of hate speech detection models, revealing crucial weaknesses in both monolingual and cross-lingual applications. Another crucial solution that was introduced in the detection of offensive and abusive language in Dutch is *Caselli & Van Der Veen (2023)*, which is a comprehensive study of fine-tuned models. The study also examines the use of data cartography to determine high-quality training data. These two mentioned studies are not only restricted to solving data bias issues, but also to evaluate pre-trained language models and LLMs, and to identify precisely the quality of datasets.

**Hallucination of LLMs:** *Bang et al. (2023)* shows that multitasking, multimodal, and multilingual use cases have profited from the usage of Large Language Models (LLMs), such as ChatGPT. Yet, in the multilingual domain, they often have issues with hallucinations. For example, the user confidence rate can be decreased by low-performant translation tools, resulting in safety problems (*Guerreiro et al., 2023*).

### Non-technical challenges

**Language cultures and dialects:** Offensive language tasks can be embedded in cultural issues. Any cultural background could impact whether a word or expression is considered offensive or not (*Schmidt & Wiegand, 2017*), thus, even utilizing the same language, this content and the capacity of offensiveness could be varied among regions and populations. Another factor is to consider the language's various dialects. For instance, the Arabic language is associated with lots of different dialects utilized by Arabic speakers on Twitter. As a result, learning and comprehending Arabic is a difficult task, especially for offensive language detection (*Al-Hassan & Al-Dossari, 2019*).

**Definition of hate speech:** As described in Section "Definition of Hate Speech", various jurisdictions give different definitions of offensive language, thus, resulting in a non-standard general definition. This becomes more complex in multilingual scenarios considering the cultural aspects and dialects.

**Annotation problem of 'foreign language effect':** The "foreign language effect" is one of the major issues of offensive data labeling, it yields people (annotators) to adopt different moral stances and usually consider this content to be less harsh in their second languages, thus affecting the multilingual annotation stability of this content. This has been studied in *Abercrombie, Hovy & Prabhakaran (2023)*, which finds out a lower annotation agreement on hateful English and German labeling tasks.

## Limitations

**Computational Limitations:** Multilingual text classification task requires extensive computational resources due to large data volume from many languages and the complexity of the models employed (since multilingual models are usually bigger in size with more weights). In fact, multilingual embeddings or pre-trained transformers (mBERT, XLM-R, mT5, among others.) require more computing resources. Moreover, the process of fine-tuning these models usually needs extensive training. While some resources provided, like Google Colab, VastAI, and cloud platforms such as AWS, Google Cloud, Microsoft Azure, and Baidu offer essential computational resources, they may be costly, especially for users and researchers with limited budgets. For further details, a brief description of the resource providers, mentioned above, is presented in Table 9, we illustrated the most affordable offers delivered.

**Multilingual Pre-trained Large Language models (Multilingual LLMs):** Implementing and training pre-trained language models is not an easy task due to their limitations (*Nozza, 2021*). An example of these crucial limitations, presented by *Conneau et al. (2020)*, is the "curse of multilinguality". They indicate that training multilingual LLMs in more languages shows declines in performance despite keeping the number of update steps. Furthermore, performance usually declines when supporting more languages and providing optimal performance on a more limited language set. This 'curse' basically involves determining whether to work on a small number of languages for more accurate performance or to distribute resources across multiple languages but with reduced performance (*Pfeiffer et al., 2022*).

**Limitations on machine translation tools:** The performance of offensive language detection models can be impacted by the quality and precision of the machine translation tools. Although multilingual Neural Machine Translation (multilingual NMT) displays significant performance, the degree to which it can handle many languages remains limited (*Aharoni, Johnson & Firat, 2019*). Recently, these tools have made important results in bilingual translation (*Cho et al., 2014*; *Vaswani et al., 2017*). However, there remain considerable barriers when it comes to implementing NMT in low-resource languages (*Dabre, Chu & Kunchukuttan, 2020*; *Wang et al., 2021*). Therefore, recent research studies have emphasized enclosing many translation data inside a single model to improve their performance (*Aharoni, Johnson & Firat, 2019*). However, it has been observed that these models frequently give low performance compared to the bilingual ones (*Arivazhagan et al., 2019*; *Pham et al., 2019*). Moreover, the majority of earlier studies have been focused on English language translation, which lead to non-English ones to be low performing (*Aharoni, Johnson & Firat, 2019*; *Zhang et al., 2020*).

## Future directions

The field of multilingual and cross-lingual offensive language detection offers many promising recommendations for future research. especially in low resource languages, as well as in different topics of offensive language. For instance, since social media users generate one-third of the poor-quality Arabic content, *Koshiry et al. (2023)* built and

**Table 9 Comparative analysis of affordable computational resources for machine learning training.**

| Platform | Resources (the cheapest offers) | Limitations |
|---|---|---|
| Google colab | Free tier provides an NVIDIA Tesla K80 GPU with 12 GB of RAM. | The session length is capped at 12 h. After this, all data will be deleted, including any trained models unless they've been saved elsewhere. |
| VastAI | Depending on demand, one can rent an NVIDIA GTX 1080 Ti with 11 GB of GPU memory for as low as around $0.10/h. | Although cost-effective, the availability of cheap resources is highly dependent on demand and can be unreliable. |
| AWS | The EC2 Spot Instances allow for cheap access to powerful resources. For instance, a g4dn.xlarge instance with an NVIDIA T4 GPU (16 GB of GPU memory) can be rented for around $0.30/h, but the exact rate varies. | Spot Instances can be interrupted by AWS with a 2-min notification. They blueare best for flexible applications that aren't sensitive to sudden interruptions. |
| Google cloud | Preemptible VMs provide affordable access to powerful resources. For example, a preemptible instance with an NVIDIA Tesla T4 GPU can be rented for approximately $0.30/h, but the exact rate varies. | Preemptible VMs can be stopped by Google at any time if resources are required elsewhere, and they automatically shut down after 24 h. |
| Microsoft azure | Azure Spot Instances provide cheaper access to resources. An example is the Standard_NV4as_v4 Spot instance with a portion (1/8) of an NVIDIA Tesla V100 GPU available for approximately $0.17/h, though exact rates vary. | Like other spot or preemptible instances, Azure Spot Instances can be interrupted by Microsoft at any time if the resources are required elsewhere. |
| Baidu cloud compute (BCC) | Offers an array of hardware resources, like the NVIDIA deep learning development card and NVIDIA Tesla K40 as cost-effective GPU for beginners and those with lower training requirements. It also offers discounts that can be checked directly in the website. | Potential language barriers given Baidu's primary focus on the Chinese market. |

annotated a standardized toxic Arabic dataset from Twitter, which would facilitate and improve toxicity analysis in Arabic language.

### On dataset

Future studies could concentrate on developing more diverse, and balanced datasets in multiple languages and dialects, as well as in different topics of offensive language.

**Generating data:** With the problem of data scarcity, especially in multilingual settings, some research studies are directed into providing more efficient solutions for data augmentation, by leveraging generated samples in order to gradually train their detection models and enhance the performance of their classification capabilities. Several approaches have been already released on English samples, that may be used to work on generating multilingual data, using different methods like adversarial auto-regressive models (*Ocampo, Cabrio & Villata, 2023*), generative GPT3 PLM-based models (*Hartvigsen et al., 2022*), or generative GPT-Neo based model (*Muti, Fernicola & Barrón-Cedeño, 2023*).

**External features:** Multiple research studies have highlighted the incorporation of features extracted from domain-agnostic or language-independent resources in cross-lingual aspects (also in cross-domain). Moreover, certain studies have underlined the vital role that emotional information has in detecting offensive language (*Rajamanickam et al., 2020*; *Safi Samghabadi et al., 2020*). Therefore, it would be helpful to examine the inclusion of emotional or sentiment data for boosting knowledge transfer. Similarly, a study performed by *Pamungkas, Basile & Patti (2021a)* confirmed the effectiveness of external features extracted from the multilingual lexicon HurtLex. They highlight its importance in

assisting the knowledge transfer process in the detection of multilingual offensive content especially when dealing with metaphors, metonymy,among others, as well as non-formal expressions that are highly sensitive to geographical, and cultural deviations.

**Advanced annotation:** Using Generative Pretrained Transformer models could increase training data (data augmentation), and despite showing promising performance in generating data in high-resource languages, this still requires to be enhanced more to generate data in low-resource languages (*Ahuja et al., 2023*). Additionally, semi-supervised learning and unsupervised methods could effectively use both labeled and unlabeled data, and reduce the challenge of annotated data scarcity.

### On modeling

Besides working on data, future studies should also focus on how to get benefit from the new innovative methods, in order to build more effective models in the field. For instance, using Federated Learning for decentralized and privacy-preserving training may assist in understanding local dialects and slang (*Weller et al., 2022*). Adding to that, Explainable AI (XAI) could guarantee more transparent model decisions, promoting trust in their classification performance (*Kumar, Dikshit & Albuquerque, 2021*). Moreover, Reinforcement learning can make models determine optimal measures for the classification task (*Fang, Li & Cohn, 2017*), and active learning can enable demanding labels for the most informative instances (*Hajmohammadi et al., 2015*). Another future aspect to be considered is the "Teacher and Student", it is used to transfer knowledge from LLMs to create smaller pre-trained models. For example, the study (*Ranasinghe & Zampieri, 2023*), worked on creating lightweight offensive language models (with fewer numbers of parameters and with less computational consumption resources), which can be among the initial steps to create multilingual models specialized more in this domain. Besides machine learning field, quantum computing has also proved to be a competitive method, faster and promising high performance in low resource languages like Arabic (*Omar & Abd El-Hafeez, 2023*).

**Study the impact of new generative models:** Future studies could explore the impact of the new generative models, such as GPT-3, GPT-4, and ChatGPT, in order to improve multilingual offensive language detection, using their ability in cross/multi-lingual understanding. They can also handle data scarcity problems by generating synthetic data in low-resource languages, much more similar to human-written data. For example, *Hartvigsen et al. (2022)* released 'ToxiGen': an English machine-generated dataset, that could be a start to create datasets in other languages in the field.

### On low resource languages

There is an increasing necessity to focus on low-resource languages, ensuring more general language coverage worldwide. For example, there are multiple research models released that worked on African languages such as Wolof and Swahili (*Jacobs et al., 2023*).

## CONCLUSION

While monolingual resources and approaches are important, the significance of multilingual efforts are highly crucial. Multilingual solutions not only broaden the scope of

understanding but also enable the development of more robust models. By addressing various languages simultaneously, we can bridge communication gaps and cultural diversity. In fact, leveraging multilingual resources promotes innovation and technological advancements, leading to more effective and universally applicable solutions. Moreover, encouraging multilingualism efforts in offensive language detection enables greater understanding, and effectiveness in safeguarding online communication. In this survey, we conduct a thorough investigation of multilingual offensive language identification in fast-globalizing social media platforms where hundreds of languages and dialects are used to communicate. Our work is motivated by the difficulty of detecting offensive content within the increasing use of non-English and low-resource languages. Our study draws inspiration from previous surveys in the field, outlining the gaps we managed to address. Specifically, our survey distinguishes itself by comprehensively presenting both multilingual and cross-lingual offensive language detection approaches, organizing findings across various machine learning classes, ranging from traditional to more advanced approaches. This inclusive strategy aims to offer readers a comprehensive understanding of existing approaches while encouraging for the adoption of more progressive techniques, detailed in the "Other Technologies" and "Future Directions" subsections. Moreover, a crucial aspect that distinguished our research is the expansive coverage of resources and tools. We prioritize the presentation of datasets, more specifically enclosing a significant number of corpora within the field, and covering a greater number of low-resource languages. We also tried to give a wider view of the other resources, getting deeply into the projects, source codes, APIs, among others. Finally, our study underlines critical challenges in the multilingual landscape of offensive language detection, attributing limitations to these issues and providing clear solutions to be considered as future directions. Overall, our survey aims to serve as a comprehensive guideline for both industry and academic practitioners, offering a significant and rich understandings into various aspects of multilingual offensive language detection while advocating for progressive advancements in the field.

### Funding
The authors received no funding for this work.

### Competing Interests
The authors declare that they have no competing interests.

### Author Contributions
- Khouloud Mnassri conceived and designed the experiments, performed the experiments, analyzed the data, performed the computation work, prepared figures and/or tables, authored or reviewed drafts of the article, and approved the final draft.
- Reza Farahbakhsh analyzed the data, prepared figures and/or tables, authored or reviewed drafts of the article, and approved the final draft.

- Razieh Chalehchaleh conceived and designed the experiments, performed the experiments, analyzed the data, prepared figures and/or tables, authored or reviewed drafts of the article, and approved the final draft.
- Praboda Rajapaksha analyzed the data, prepared figures and/or tables, authored or reviewed drafts of the article, and approved the final draft.
- Amir Reza Jafari analyzed the data, authored or reviewed drafts of the article, and approved the final draft.
- Guanlin Li conceived and designed the experiments, performed the experiments, analyzed the data, authored or reviewed drafts of the article, and approved the final draft.
- Noel Crespi analyzed the data, authored or reviewed drafts of the article, and approved the final draft.

## Data Availability

All relevant of this manuscript, including the elements we mentioned in our survey (existing studies, datasets, and resources) are available (and will be completed and actively maintained) at GitHub: https://github.com/KhouloudMN97/A-Survey-on-Multi-lingual-Offensive-Language-Detection/tree/main.

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
