# Peer review of "A survey on multi-lingual offensive language detection"

_PeerJ Computer Science, doi:10.7717/peerj-cs.1934_

## Round 0.1 · original submission · Major Revisions

Given that all three reviewers require many changes, the paper needs to be revised with details of changes provided if the authors submits the changes.

**Language Note:** The review process has identified that the English language must be improved. PeerJ can provide language editing services - please contact us at copyediting@peerj.com for pricing (be sure to provide your manuscript number and title). Alternatively, you should make your own arrangements to improve the language quality and provide details in your response letter. – PeerJ Staff

·

Basic reporting

1. In Table 1. Key previous surveys on the topic of (multilingual) hate speech detection.( How the study published in 2021 and authors set year in 2023 in rows 4 and 6)
2. In each table add the metrics of each study (accuracy, Precision, Recall, F1 score) if found.
3. The authors need to make a clear proofread to avoid grammatical mistakes and typo errors.
4. Add future work in last section (conclusion) (if any)
5. What are the recommendations of this review.
6. To improve the Related Work and Introduction sections authors are recommended to review this highly related research work paper:
a) The effect of rebalancing techniques on the classification performance in cyberbullying datasets
b) Arabic Toxic Tweet Classification: Leveraging the AraBERT Model
c) Quantum computing and machine learning for Arabic language sentiment classification in social media
d) Comparative performance of ensemble machine learning for Arabic cyberbullying and offensive language detection
e) Automatic detection of cyberbullying and abusive language in Arabic content on social networks: a survey
f) User Awareness of Privacy, Reporting System and Cyberbullying on Facebook
g) Comparative performance of machine learning and deep learning algorithms for Arabic hate speech detection in osns

Experimental design

Article content is within the Aims and Scope of the journal.
Rigorous investigation performed to a high technical & ethical standard.
Methods described with sufficient detail & information to replicate.

Validity of the findings

Impact and novelty not assessed. Meaningful replication encouraged where rationale & benefit to literature is clearly stated.
Conclusions are well stated, linked to original research question & limited to supporting results.

Is there a well developed and supported argument that meets the goals set out in the Introduction?
Does the Conclusion identify unresolved questions / gaps / future directions?

All of above need to be improved.

Cite this review as

·

Basic reporting

The paper targets "multilingual offensive language". The problem I have is that the paper covers more than this and it fails to maintain a coherent use of the terminology. In the Introduction, the authors shift from offensive language to hate speech, toxic language and abusive language (all in the very beginning - lines 23-28) as if these are all synonyms rather than different phenomena. As the authors acknowledge in Figure 1, phenomena differ for intentions, targets, and impact (on targets). I think the review will benefit if, already in the introduction, these terms are presented as being strictly connected but different phenomena.

There have been some recent reviews in the filed. Among those, the one that has not been mentioned is Vidgen and Derczynski (https://arxiv.org/pdf/2004.01670v3.pdf) - which I think should be listed and discussed.

I am not fully satisfied by this version of the paper for multiple reasons:

a) the work is poorly focused: I think the authors fail to actively differentiate from other existing work - especially Poletto et al., (2021) and Jahan and Oussalah (2023; now available here: https://www.sciencedirect.com/science/article/pii/S0925231223003557#s0115). If the topic is "multilingual offensive language" then focus only on multilingualism: of datasets and approaches. In your work you mix monolingual and multilingual in too many cases. I would find interesting to read a survey that discuss - more in depth that what has been done in this version - issues related to multilingualism for offensive language and related phenomena. A key element - in my understanding of such a survey - is that one wants to focus on datasets in different languages that use the same definition of the targeted phenomena or coordinated efforts across of monolingual corpora (still sharing the same definition).

b) when it comes to the systems that have been developed for addressing the multilingualism of offensive language (here used as a broad umbrella term), I would have appreciated a more in-depth discussion of the methods used, what is working, what is not working, and how they compare with respect to monolingual approaches.

c) the target audience is missing - if this is directed at experts then it is too high level to be of interest

The language used is mostly clear and easy to follow, but some passages needs to be revised and/or improved:
- line 25: platforms was an important --> platforms is an important
- lines 83-86: what do you refer to when stating "hate speech content in social media refers to any content posted on social media platforms that attack individuals or groups based on race, ethnicity, religion, sexual orientation, gender, or any other characteristic." Are you referring to an analysis of multiple terms of use statements of platforms? on what is this claim based?
- lines 107-108 "multilingual hate speech and offensive language detection.": not clear; I think you can break this down in the components you are reviewing: datasets and systems for multilingual hate speech and offensive language detection
- line 118 "provided" ? not clear: delete the word or rephrase the sentence
- line 130 "general overview of hate speech and offensive language detection" it is not clear. Do you mean: an overview of "the approaches for [...]"?
- line 144 "metric," --> aspect + remove the comma
- line 185: similar comments for lines 130 and 107-108
- line 192: multilingual offensive --> what? something is missing; e.g. "offensive language from a multilingual perspective" ?
- line 479 Following Twitter --> After Twitter
- line 490 "highly targeted target groups" --> highly targeted groups
- line 493 "Negative speech datasets" : this term has never been used before. Be consistent. It could be anticipated and used an umbrella term for all phenomena you want to cover - for instance
- line 498: impressed: ? not clear what you mean with this word here
- line 505 ", offering valuable insights for researchers in the field." personally, I think these sentences should be avoided - let the readers judge. If you want to use it, then point out why what you have presented is "valuable"
- line 525 - 526 "Please Note" --> Please note
- lines 638 - 649: the whole paragraph reads as a repetition of what discussed between lines 613 - 637. Either you delete this part or you make it different
- lines 665 - 666: In 2020 "Detoxify" project [...] --> the sentences does not seem grammatical - please adjust.
- line 687 "As well as openai --> maybe start with "Similarly, openai [...]
- line 690: "the significance" --> why "significance"? not clear what is meant here. Maybe the "relevance" ?

When it comes to the style, the followings are points of improvement to be addressed:
- be precise: do not use "etc" or suspension points. To close a potentially very long list, you can use the words "among others".
- references: all references are directly in the text body. References should be in parentheses except when the author(s) are referred directly. Use the Latex command \cite{} or \citep{}.
- do not use abbreviations (e.g. they're)

Experimental design

The approach used to conduct this survey is not systematic enough and not fully comprehensive. More systematicity in selecting the work that are discussed in this review would have made the work more solid. Why the authors did not start from the Hate Speech Dataset catalogue (https://github.com/leondz/hatespeechdata)?

As I said above, the lack of systematicity is also reflected on the continuous shifts of phenomena that are covered. The title is on "offensive language" - which is a phenomenon on itself. You then cover toxicity and hate speech. This is not per se a problem. The problem is that it seems that all these different phenomena are the same, while it is not the case.

It is highly unclear why some monolingual corpora are present and other are excluded (my suggestion: you want to exclude all monolingual corpora and focus only on the multilingual ones).

One aspect that is not discussed at all concerns the subjectivity of the annotations. In your overview of the benchmark, there is no information on how they have been collected and annotated, nor of their representativeness of the phenomena as well as of potential bias. This is an extremely hot topic in current debates on offensive language phenomena.

I would re-organize the flow of information as follows:
- first, discuss the phenomena that are covered when using the term "offensive language, clarify what is and what is out; clarify your terminology (e.g., negative language as a cover term and more specific terms for the various phenomena)
- present the methodology of your review: this should be discussed in detail and compared with previous work (how is your review different from the ones that are very recent?)
- discuss the benchmark data - I would also include a discussion on the dynamic and functional benchmarks that have been developed recently (e.g. https://aclanthology.org/2022.woah-1.15/; https://aclanthology.org/2023.woah-1.7.pdf)
- present the approaches: focus only on multilingual approaches. Be precise on the time period that you cover. I would also discuss the results of the systems. I think that the available APIs should be discussed in the section rather then in the "Resouce" section
- Cover shared task
- Cover previous projects. Again: be more systematic in your analysis. There are lots of projects that are missing and it is difficult to understand why. Some examples of projects not covered: https://www.textgain.com/projects/eooh/; https://www.standbymeproject.eu ; http://hatemeter.eu - among others).

Validity of the findings

My main issue here s that I am not able to answer the following questions clearly: "What did I learn from this review?" "What are the gaps to be addressed?".

The discussion in the last section on the challenges and limitations is limited. There are many questions and issues that are not even discussed:
- is it worthwhile to do multilingual offensive language or shall we focus on monolingual resources and approaches?
- how can multilingual approaches be improved?
- how can we create datasets that are more suitable for these tasks?
- how about the subjectivity and the context of occurrence of the message?
- How about multimodality? Can we completely avoid it?
- How about the definitions?

Definitions of different phenomena is not a minor aspect in your survey, I think. Since you include monolingual resources, can they be compared one to each other? Do they adopt the same definitions?

Additional comments

I think that the idea of the survey is good but it really needs to be narrowed and more focused.

Figure 1: discuss how you have come to this figure. It presents different phenomena and their relationships - I am not sure that Homophobia can qualify as "aggressive content" - it is an instance of hate speech. How was this Figure designed? How your dimensions and instances where identified and put in relations? How do you differ from Poletto et al., 2021?

line 364-365 "Pamungkas and Patti (2019) implemented a domain-independent lexicon called “HurtLex” in cross-domain and cross-language methods [...]" This is not correct. Pamungkas and Patti (2019) used HurtLex (correct citation is Bassignana et al., 2018 - https://ceur-ws.org/Vol-2253/paper49.pdf). Please revise the paper and correct this sentence.

Cite this review as

·

Basic reporting

No comment

Experimental design

no comment

Validity of the findings

no comment

Additional comments

This manuscript is a survey paper about multilingual offensive language detection. Topic is really very interested and unique in its presentation and contents. Selection of different topics and content presentation is good. I have the following observations about the manuscript:
1. It is good to include a section with a table or a graph to show the different repositories from the different papers has been collected. For examples, Elsevier, Springer, IEEE or Q1, Q2, Q3 journals… conference and journal papers. This representation will attract more readers by developing their interest in this study.
2. I also suggest to add a table in the availabale datasets section where the publically available datasets should be organized by their size and classes. This will help the readers to make a quick decision about the choice of the dataset for next study.
3. In Section “ Approaches on Multilingual Hate Speech Detection”, Reinforcement learning and Transformer based studies have not been included. I suggest to add these two as a subsections.
4. I suggest to add some more graphics than tables.

Cite this review as

---

## Round 0.2 · accepted · Accept

Thanks for undertaking the major review of the paper. The current version is acceptable to the reviewers.

·

Basic reporting

Accept.

Experimental design

-

Validity of the findings

-

Cite this review as